# Robust ultrathin nanoporous MOF membrane with intra-crystalline defects for fast water transport

Xueling Wang[1], Qiang Lyu [2], Tiezheng Tong [3], Kuo Sun[1], Li-Chiang Lin[2], Chuyang Y. Tang[4], Fenglin Yang[1], Michael D. Guiver [5✉], Xie Quan [1✉] & Yingchao Dong [1✉]

Rational design of high-performance stable metal–organic framework (MOF) membranes is challenging, especially for the sustainable treatment of hypersaline waters to address critical global environmental issues. Herein, a molecular-level intra-crystalline defect strategy combined with a selective layer thinning protocol is proposed to fabricate robust ultrathin missing-linker UiO-66 (ML-UiO-66) membrane to enable fast water permeation. Besides almost complete salt rejection, high and stable water flux is achieved even under long-term pervaporation operation in hash environments, which effectively addresses challenging stability issues. Then, detailed structural characterizations are employed to identify the type, chemical functionality, and density of intra-crystalline missing-linker defects. Moreover, molecular dynamics simulations shed light on the positive atomistic role of these defects, which are responsible for substantially enhancing structural hydrophilicity and enlarging pore window, consequently allowing ultra-fast water transport via a lower-energy-barrier pathway across three-dimensional sub-nanochannels during pervaporation. Unlike common unfavorable defect effects, the present positive intra-crystalline defect engineering concept at the molecular level is expected to pave a promising way toward not only rational design of next-generation MOF membranes with enhanced permeation performance, but additional water treatment applications.

[1] Key Laboratory of Industrial Ecology and Environmental Engineering (Ministry of Education, MOE), School of Environmental Science and Technology, Dalian University of Technology, Liaoning 116024 Dalian, China. [2] William G. Lowrie Department of Chemical and Biomolecular Engineering, The Ohio State University, Columbus, OH 43210, USA. [3] Department of Civil and Environmental Engineering, Colorado State University, Fort Collins, CO 80523, USA. [4] Department of Civil Engineering, The University of Hong Kong, Pokfulam, Hong Kong, China. [5] State Key Laboratory of Engines, and Collaborative Innovation Center of Chemical Science and Engineering (Tianjin), Tianjin University, Tianjin 300072, China. ✉email: michael.guiver@outlook.com; quanxie@dlut.edu.cn; ycdong@dlut.edu.cn

Water scarcity has become a global challenge due to increasing water demand and severe water pollution[1–3]. To mitigate this, seawater desalination and wastewater recycling have been being increasingly utilized for large-scale clean water supplies[1,4]. Reverse osmosis (RO) is the most widely used membrane-based desalination technology[5]. However, it is unsuitable for high salinity water sources with ultrahigh osmotic pressure[6–8]. Current RO membranes also suffer from unsatisfactory chlorine resistance. Although conventional thermal methods, such as multi-effect distillation (MED) and multistage flash distillation (MSFD), are capable of treating high salinity wastewaters[9], they require high energy consumption, extensive infrastructure and footprint, and high capital cost[10,11]. In contrast, hybrid thermal-membrane processes such as membrane distillation (MD) are increasingly investigated to effectively desalinate hypersaline feeds due to their excellent separation efficiency and ultrahigh salt enrichment ability[12]. However, due to the presence of various detrimental species in real waters, membrane fouling, scaling, and wetting often result in performance degradation of MD, especially over long-term operation[13–15]. These shortcomings motivate us to develop alternative technologies.

Pervaporation (PV) is an alternative process enabling more challenging separations of various azeotropes and water–alcohol mixtures[16,17]. In the PV process, liquid water molecules can transport across a dense polymeric (or nanoporous inorganic) membrane selective skin layer via a dissolution–diffusion (or adsorption–diffusion) mechanistic process followed by evaporation–condensation even at room temperature, while salt ions are rejected[18,19]. A key challenge in PV desalination is the lack of highly water-permeable stable membranes. In spite of satisfying salt rejection (99%), most current state-of-the-art PV membranes, such as zeolite, suffer from very low water flux only ranging from 1.0 to 4.0 L m$^{-2}$ h$^{-1}$ due to their unfavorable micrometer level thickness of the selective layer[16,20,21]. In contrast, metal-organic framework (MOF) membranes have received attention due to their high porosity, well-defined pore architecture, designable structures, and wide diversity in both cluster and linker[22–24]. Most MOFs are not particularly stable in aqueous media applications such as membrane separation. Despite their enhanced flux compared with zeolite membranes, current water-stable MOF membranes exhibit moderate flux (most <10 L m$^{-2}$ h$^{-1}$)[25,26], which needs to be further improved for future large-scale applications. To address this, the development of ultrathin stable MOF membranes would be feasible to improve water flux due to much lower transport resistance, but is experimentally difficult[27–29]. Beyond this, further enhancing water flux is more challenging, which calls for the rational design of MOF membrane structures at the molecular or atomic level. Defects in most separation membranes usually play a negative role in performance, especially selectivity deterioration[30,31]. However, the manipulation of defects to improve permeation performance is uncommon. Existing strategies of defect chemistry control in MOF crystals[32–34] provide us with some inspiration to rationally design MOF membranes with favorable intra-crystalline defects to enhance permeation performance, without compromising selectivity. Especially, modulation of intra-crystalline missing-linker defects is considered as a feasible method using carboxylic-acid as modulator since it can coordinate with metal clusters by competing with carboxylic-bearing linkers[35,36].

In this work, besides implementing a facile thinning protocol, a molecular-level intra-crystalline defect chemistry strategy is proposed to rationally design a robust ultrathin missing-linker UiO-66 (ML-UiO-66) membrane to substantially improve water permeation during the PV process (Fig. 1). Ultrathin ML-UiO-66 membranes are demonstrated to have not only excellent stability toward hot saline, chlorine, alkaline and acidic solutions, but also almost complete salt rejection and more importantly high water flux (~29.8 L m$^{-2}$ h$^{-1}$), outperforming existing state-of-the-art zeolite and MOF PV membranes. Furthermore, for both low- and high-concentration saline waters, they also show long-term operating stability. To further reveal molecular-level insights into defect-enhanced performance, both structural characterization and molecular dynamics simulations are carried out to shed light on the atomic role of missing-linker defect chemistry in the ML-UiO-66 membranes. Our work provides a promising approach toward molecular-level design of highly permeable MOF membranes for challenging separation (e.g., harsh hypersaline water desalination) and a fundamental understanding of how introducing deliberate defects enhances permeation performance.

## Results

**Construction of robust ultrathin ML-UiO-66 membranes**. ML-UiO-66 exhibited excellent structural stability toward hot highly saline solutions (Fig. 2a). Moreover, its phase structure still remained unchanged after long-term immersion into diluted alkaline, strongly acidic, and even chlorine-containing (NaClO) saline solutions (Fig. 2b). These characteristics indicate that ML-UiO-66 has promising potential in harsh desalination applications, due to its strong coordination bonds between [Zr$_6$(OH)$_4$O$_4$] cluster and monocarboxylate/dicarboxylate as well as robust Zr–O bonds[37]. In addition, beyond commercial flat-sheet/tubular and other hollow fiber substrates, macro-porous ZrO$_2$ substrates (Supplementary Fig. 1) and nano-interlayer modified ZrO$_2$ substrates in this work (Supplementary Figs. 3–9) have a much higher packing density (~4340 m$^2$ m$^{-3}$, 5–86 times higher than commercial ceramic substrates)[38], which is favorable to provide higher treatment efficiency using size-minimized membrane modules (Fig. 2c). However, on such coarse ceramic substrates (pore size ~142 ± 48 nm) (Supplementary Fig. 5), the as-grown ML-UiO-66 membranes were too thick (in the micrometer range) with poor inter-growth quality (e.g., large inter-crystalline defects) (Fig. 2g, h and Supplementary Fig. 10), inevitably resulting in poor separation performance especially selectivity (Supplementary Fig. 13). This can be ascribed to the coarse substrates (sintered at ~1100 °C) only providing low-density heterogeneous nucleation sites (i.e., –OH group) (Supplementary Fig. 6) and high surface roughness ($R_a$ = ~27 ± 3 nm) (Supplementary Fig. 7), resulting in the formation of a rough ($R_a$ = ~133 ± 2 nm) thick layer with poorly inter-grown large ML-UiO-66 crystals (Fig. 2g, h and Supplementary Fig. 12)[39]. To address this issue, a substrate surface engineering protocol was employed by introducing a tailored nanoporous γ-Al$_2$O$_3$ interlayer (pore size ~48 ± 1 nm) (Supplementary Fig. 5) onto the potentially scalable ceramic substrates by the conventional dip-coating method using boehmite sol (Supplementary Figs. 3–4). After sintering at 750 °C, the resulting low-roughness ($R_a$ = ~10 ± 2 nm) interlayer (Supplementary Fig. 7) provided high-density nucleation sites with abundant surface –OH groups (Supplementary Fig. 6), facilitating the growth of ultrathin ML-UiO-66 membranes with a thickness of only 103 ± 14 nm, which is much thinner (1/8–1/35) than reported state-of-the-art UiO-66 membranes (Fig. 2d–f and Supplementary Table 3)[30,31]. Besides such significantly reduced thickness, ML-UiO-66 membranes also exhibited a high-quality compact membrane morphology with well inter-grown smaller crystals and lower surface roughness ($R_a$ = ~47 ± 27 nm) (Fig. 2d–f, Supplementary Figs. 11, 12, and 14 and Supplementary Table 4). In spite of introducing missing-linker intra-crystalline defects, ML-UiO-66 membranes still show

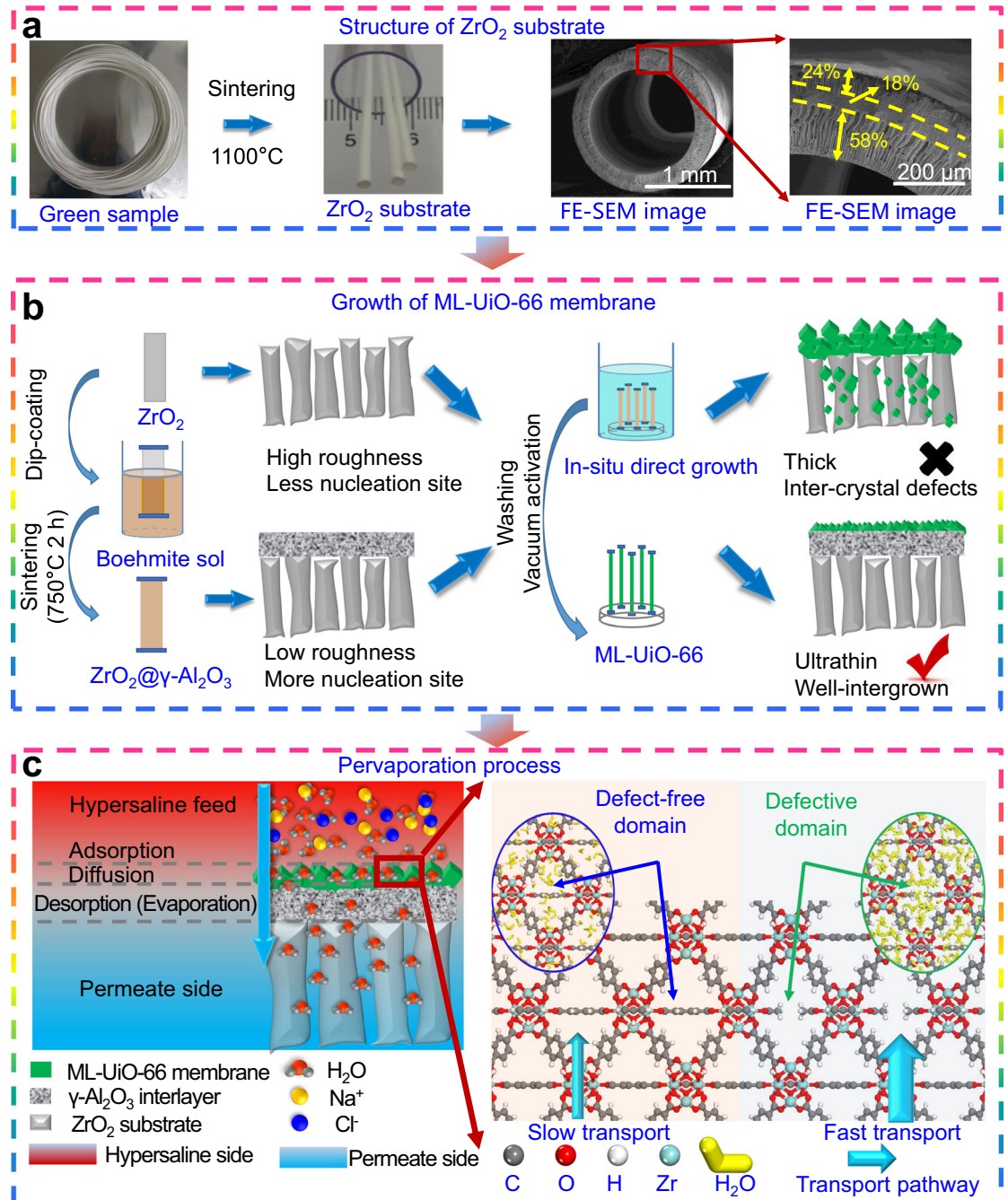

**Fig. 1 Schematic diagram of fabrication procedures and desalination mechanistic process of ultrathin ML-UiO-66 membranes via introducing γ-Al$_2$O$_3$ interlayer onto scalable coarse ZrO$_2$ ceramic substrates (denoted as ZrO$_2$@γ-Al$_2$O$_3$ hereafter) (Supplementary Figs. 1, 2). a** Structure of ZrO$_2$ ceramic substrate (digital photographs, cross-sectional filed emission scanning electron microscopy (FE-SEM) image and locally enlarged cross-sectional FE-SEM image). **b** Fabrication of γ-Al$_2$O$_3$ interlayer and growth of ML-UiO-66 membrane. **c** Pervaporation desalination process (left) and mechanism of intra-crystalline defect-enhanced water permeation (right).

a typically pure UiO-66 phase (Fig. 2i). Meanwhile, the average pore size was enhanced due to the introduction of intra-crystalline defects (0.586 nm for UiO-66 membrane and 0.628 nm for ML-UiO-66 membrane) (Supplementary Fig. 15).

**High-performance harsh desalination.** Most previous studies have largely focused on RO desalination performance of UiO-66 membranes. Due to their micrometer thickness range of the selective layer, the reported UiO-66 membranes had very low water flux and permeability (i.e., only 0.14–1.5 L m$^{-2}$ h$^{-1}$ bar$^{-1}$ for RO)[31,40]. More importantly, they suffered from poor rejection,

especially for monovalent salt (e.g., <50% for NaCl), severely limiting their utility (Supplementary Table 5)[31,40]. Only one report involved PV desalination using UiO-66 membrane[26]. In spite of satisfying NaCl rejection, its flux was only moderate, which needs to be improved. In comparison with state-of-the-art MOF membranes almost in the micrometer thickness range, the ML-UiO-66 membrane exhibits characteristics consistent with an ultrathin selective layer of only 103 ± 14 nm (Fig. 3c). Specially, the ML-UiO-66 membrane had a much higher water flux than other reported zeolite and MOF membranes (~104–298 times higher by the RO process (at 1.0 bar) and ~2.2–85 times higher by the PV

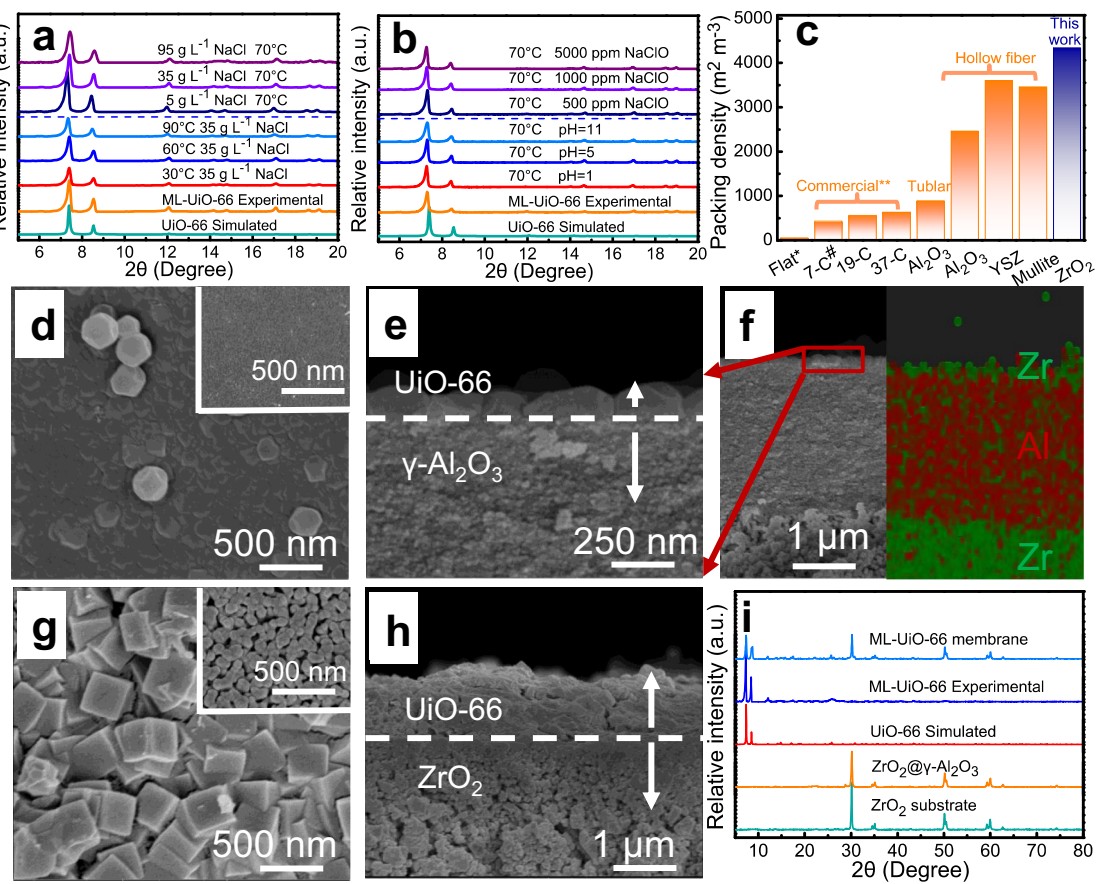

**Fig. 2 Stability and structure characterization. a** X-ray diffraction (XRD) patterns of ML-UiO-66 before and after 10-day long-term immersion into saline solutions at different temperatures (30, 60, and 90 °C) and different salt concentrations (5, 35, and 95 g L$^{-1}$). **b** XRD patterns of ML-UiO-66 after 10-day immersion into aqueous salt solutions with different pH values (1, 5, and 11) and different NaClO concentrations (500, 1000, and 5000 ppm). **c** Comparison of packing density between ZrO$_2$ substrate in this work and other ceramic substrates[60–62] (*Shandong Industrial Ceramic Research & Design Institute Co., Ltd., China, **Great Wall Xinyuan Membrane Technology Co., Ltd., China, # C refers to the number of channels for commercial tubular multi-channel alumina ceramic membranes). **d** Surface field emission scanning electronic microscopy (FE-SEM), (**e**) high-magnification cross-sectional image and (**f**) cross-sectional energy dispersive spectroscopy (EDS) image (zirconium element: green signal; aluminum element: red signal) of ultrathin ML-UiO-66 membrane after in situ direct growth on lower-roughness ZrO$_2$@γ-Al$_2$O$_3$ substrate with much finer pores (the inset of (**d**)). **g** Surface FE-SEM and (**h**) cross-sectional images of thick ML-UiO-66 membrane with large inter-crystalline defects after in situ direct growth on macro-porous ZrO$_2$ substrate (the inset of (**g**)). **i** XRD patterns of ultrathin ML-UiO-66 membrane (on ZrO$_2$@γ-Al$_2$O$_3$ substrate), ML-UiO-66 powder, UiO-66 (calculated), ZrO$_2$@γ-Al$_2$O$_3$ substrate, and ZrO$_2$ substrate.

process) (Fig. 3a–c, Supplementary Fig. 16). In particular, even at the similar thickness (103 ± 14 nm vs. 120 ± 20 nm), the water flux of ML-UiO-66 membrane was enhanced by more than 49.0% compared with the UiO-66 membrane (Supplementary Fig. 17 and Supplementary Table 6). Given the same membrane thickness with continuous inter-crystalline morphology, such an enhancement in water flux can be ascribed to the difference of their intra-crystalline structure (i.e., missing-linker defects, which will be discussed later). Moreover, the introduction of missing-linker defects efficiently enhances surface hydrophilicity (Supplementary Fig. 21), water adsorption (Supplementary Fig. 22) and diffusivity ability even at room temperature (25 °C) (Supplementary Table 7), which could be further enhanced at higher temperature under PV conditions (70 °C). With increasing operation temperature from 30 to 90 °C, the water flux of ML-UiO-66 membrane increased from 9.1 ± 0.7 to 45.1 ± 1.6 L m$^{-2}$ h$^{-1}$ due to an enhanced driving force for faster water molecular diffusion transport within the 3D sub-nanometer channels[26], while maintaining stable salt rejection of >99.8% (Fig. 3d). Here, the water transport process is dominated by a typical thermally activated diffusion mechanism where the activation energy ($E_a$) is about 24.8 kJ mol$^{-1}$ (Fig. 3d inset),

which is lower than most MD or PV processes (Supplementary Table 8). Even at 30 °C (close to room temperature), a high water flux of 9.1 ± 0.7 L m$^{-2}$ h$^{-1}$ was achieved with high rejection (~99.9%) for 35 g L$^{-1}$ NaCl solution. With increasing feed salt concentration, water flux gradually decreased at the same temperature due to the adsorption–diffusion of less water molecules across the membranes with lowered driving force, since liquid water molecules need to overcome a higher energy barrier in high salinity solutions (more hydrated Na$^+$ and Cl$^-$ ions). Nevertheless, salt rejection was maintained at a consistently stable level (> 99.8%) even for ultrahigh salinity feed (95 g L$^{-1}$) (Fig. 3e), while water flux still remained a high level (16.3 ± 0.3 L m$^{-2}$ h$^{-1}$), indicating its promising potential for hypersaline water treatment[26]. The flux performance is highly competitive when compared with current state-of-the-art inorganic MD membranes, even when operating at a lower salinity level (~35 g L$^{-1}$) (Supplementary Table 9).

Long-term operational stability is an important performance indicator for real applications where most MOF membranes suffer from poor performance, especially under harsh conditions such as long-term operation, high temperature, and high

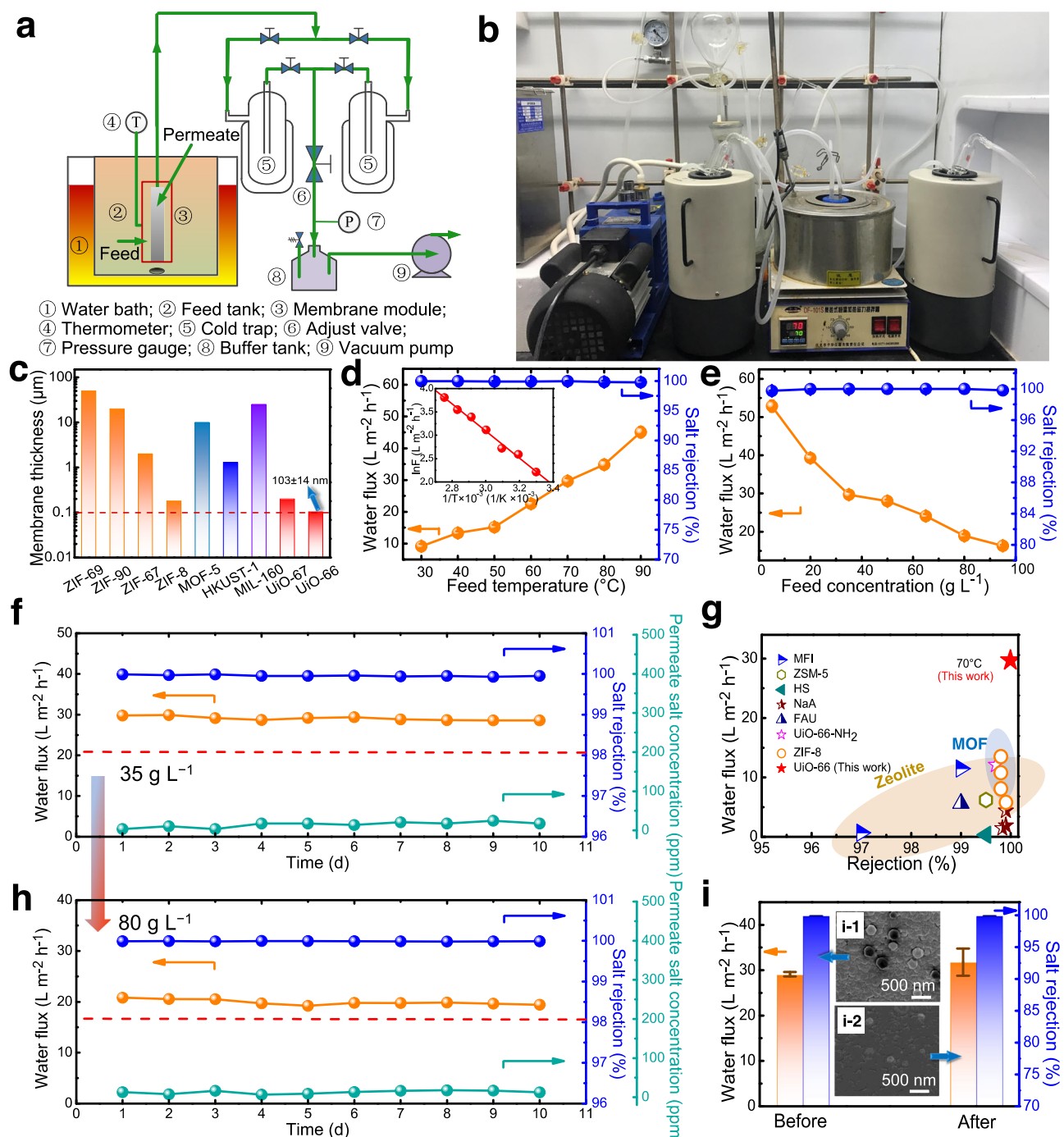

**Fig. 3 Desalination performance of robust ultrathin UiO-66/ML-UiO-66 membranes. a** Schematic diagram and (**b**) photograph of a PV setup for testing UiO-66/ML-UiO-66 membranes fabricated in this work. **c** Comparison of membrane thickness via a solution-based solvothermal method between UiO-66 membrane fabricated in the current study and other MOF membranes with lowest thickness reported in the literature[63-70]. **d** Water flux and salt rejection as a function of feed temperature during the PV process (35 g L$^{-1}$ NaCl solution) (the inset shows an Arrhenius plot between water flux and reciprocal of feed absolute temperature). **e** Water flux and salt rejection as a function of feed salt concentration ranging from 5 g L$^{-1}$ to 95 g L$^{-1}$ at a constant feed temperature of 70 °C. **f** Long-term operating desalination performance (water flux, salt rejection and permeate salt concentration) for the treatment of 35 g L$^{-1}$ NaCl solution for 240 h (10 days) at 70 °C without cleaning. (The red dashed lines in Fig. 3f, h represent the limit (Na$^+$ < 200 ppm) of drinking-water quality (fourth edition) issued by the World Health Organization (WHO)). **g** Comparison of water flux and rejection between our ultrathin ML-UiO-66 membrane and other state-of-the-art polycrystalline zeolite and MOF membranes via PV process (See details in Supplementary Table 9). **h** Long-term operating desalination performance (water flux, salt rejection and permeate salt concentration) for the treatment of hypersaline water (80 g L$^{-1}$ NaCl solution) for 240 h (10 days) at 70 °C without cleaning. **i** Water flux and salt rejection of ultrathin ML-UiO-66 membranes before and after immersion into NaClO solution (1000 ppm) for 240 h (10 days) for an accelerated chlorine resistance test (the inset shows the FE-SEM images of ML-UiO-66 membrane surface before (**i-1**) and after (**i-2**) the chlorine resistance test). (PV conditions: feed 35 g L$^{-1}$ NaCl solution, temperature: 70 °C). (Numbers that follows the ±signs are standard deviation (SD) in this study.).

salinity[24]. Interestingly, like UiO-66 membrane (Supplementary Fig. 18), ultrathin ML-UiO-66 membrane exhibited excellent 10-day stability with stable water flux (29.8–28.6 L m$^{-2}$ h$^{-1}$) and salt rejection (~99.9%) for treatment of 35 g L$^{-1}$ NaCl solution at 70 °C (Fig. 3f). The salt concentrations in the permeates were in the range of 3.5–24.5 ppm, which are far below the limit (Na$^+$ < 200 ppm) of drinking-water quality (fourth edition) issued by the World Health Organization (WHO)[41,42]. Both the surface morphology and phase structure of the ML-UiO-66 membrane were maintained without apparent changes after the 10-day stability test, indicating the excellent stability (Supplementary Fig. 23). Moreover, besides excellent salt rejection, our ML-UiO-66 membrane exhibited a high water flux (29.8 L m$^{-2}$ h$^{-1}$), outperforming other state-of-the-art polycrystalline zeolite and MOF membranes (Fig. 3g, Supplementary Table 9). Nevertheless, it is moderate when compared with other types of membranes such as PVA and graphene oxide (via PV process)[43], and poly(vinylidene fluoride) (via MD process)[44] due to the different nature of materials and membrane structures. Even for hypersaline water (80 g L$^{-1}$ NaCl solution), the ML-UiO-66 membrane exhibited not only high flux (20.8 L m$^{-2}$ h$^{-1}$) and salt rejection (~99.9%), but also excellent long-term stability (like UiO-66 membrane, Supplementary Fig. 19), with almost constant flux and rejection over a 10-day continuous operating period, further indicating its promising potential for hypersaline water treatment (Fig. 3h).

One of the big challenges for polymeric desalination membranes such as RO is their weak chlorine resistance, usually resulting in severe performance degradation[6]. Interestingly, even after immersion into NaClO aqueous solution (1000 ppm) for 10 days (equivalent to 240,000 ppm h chlorine exposure), both UiO-66 and ML-UiO-66 membranes exhibited almost unchanged membrane surface morphology and water flux, and particularly salt rejection for 35 g L$^{-1}$ NaCl solution, indicating much better chlorine resistance than polymeric counterparts (Fig. 3i and Supplementary Fig. 20). This implies that a ML-UiO-66 membrane may be expected to maintain serviceable water flux and salt rejection for as long as 6.8–13.7 years if a typical dosage of NaClO ~ 2–4 ppm is applied[6,45]. Its excellent chlorine resistance is expected not only to address biofouling in chlorine-disinfectant environment but also to improve the operational reliability and practical feasibility in actual water treatment processes.

**Identification and quantification of missing-linker defects.** Both experimental characterization (such as XRD, TG, BET, ATR-FTIR, and NMR) and computational simulations (such as molecular dynamics simulations and structural analyses) were systematically performed to reveal molecular-level mechanistic insights into the defect-enhanced permeation performance of ML-UiO-66 membranes during the PV process. Compared to defect-free UiO-66, the existence of certain defects in the ML-UiO-66 structure is confirmed by increased specific surface area and lower linker weight loss based on BET and TG analysis (Supplementary Figs. 24, 25, Supplementary Table 10). Generally, there are two types of intra-crystalline defects, which are missing-linker (Fig. 4g) and missing-cluster, in defective UiO-66 MOF structures. Compared with defect-free UiO-66 structure, a new broad XRD diffraction reflection is present in the 2θ range of 3°−7° only for missing-cluster defective UiO-66 structure due to the presence of **reo** phase[46]. In contrast, there is no such a characteristic diffraction reflection for both missing-linker defective and defect-free UiO-66 structures. Therefore, in our work, missing-cluster defects can be excluded according to the low-angle experimental and simulated XRD patterns in which

there is no broad diffraction reflection in the 2θ range of 3°−7° (Fig. 4a)[47,48]. In summary, we can reasonably infer that the defects in ML-UiO-66 membrane should be missing-linker defects. In a defect-free UiO-66 structure, each BDC$^{2-}$ linker coordinates with two Zr$_6$ clusters. In contrast, for a missing-linker defect structure, the absence of one BDC linker with negative charge causes the formation of two defect centers having four coordinated unsaturated Zr sites[35].

Besides defect-type identification, the density of missing-linker defect was also quantified based on the results of BET, TG, and theoretical surface area calculations (Fig. 4b, c). Specially, Zeo++ was employed to compute the surface area of ideal missing-linker structures and the results showed that the specific surface area linearly increases with defect density (Fig. 4b)[49]. Combined with the BET results, a missing-linker defect density of ~24.4% was determined for ML-UiO-66 (Fig. 4b), which is in agreement with that (~23.5%) calculated from the TG curves (see details on Supplementary Page S44). Using the same protocols, the missing-linker defect densities of defect-free UiO-66 were experimentally determined to be only ~0.17% (BET, Supplementary Fig. 24) and ~1.09% (TG, Supplementary Fig. 25). Considering experimental errors, these values (~0.17%, ~1.09%) are indeed close to the theoretical value of zero, indicating its nearly defect-free nature, as expected. Interestingly, due to the introduction of missing-linker defects, an enhancement in specific surface area was confirmed in the ML-UiO-66 structure (~1249.0 m$^2$ g$^{-1}$), which is ~26% higher than that (~990.4 m$^2$ g$^{-1}$) in defect-free UiO-66 (Supplementary Fig. 24, Supplementary Table 10), consequently indicating the more porous nature of ML-UiO-66 and thus the possibility of enhanced water adsorption–diffusion transport[50–52]. To further reveal the defect chemistry, compensating ligands were further identified in the ML-UiO-66 membrane structure. Usually, the terminal group in defective UiO-66 is monocarboxylate, chloride, or hydroxy group, depending on the type of modulator[48,50,51]. A broad band centered at 3450 cm$^{-1}$ is due to the presence of intercrystallite water or/and physisorbed water (ATR-FTIR spectra, Fig. 4e)[53]. The carboxylate groups show characteristic bands at 1654 (C=O stretching vibration), 1578 (C=O antisymmetric stretching vibration) and 1398 cm$^{-1}$ (C=O symmetric stretching vibration). The absence of additional hydroxy adsorption bands indicates that no hydroxy compensating ligand was formed under synthesis conditions where moisture did not interfere with the coordination reaction. Without the presence of chlorine element (EDS spectra, Fig. 4d) and additional hydroxy adsorption bands (ATR-FTIR spectra, Fig. 4e), we can conclude that the intra-crystalline defects in the ML-UiO-66 membrane structure are not terminated by chloride ligands or hydroxy groups[50]. Moreover, the liquid $^1$H NMR spectrum indicates that the defect-compensating group is indeed monocarboxylate (Fig. 4f). In conclusion, these detailed characterizations provide strong evidence that the introduction of CH$_3$COOH effectively created missing-linker defects in the UiO-66 intra-crystalline structure (i.e., ML-UiO-66) (Fig. 4g and Supplementary Fig. 26).

**Mechanistic insight into defect-enhanced performance.** To shed light on the effect of missing-linker defects on the PV desalination performance, molecular dynamics simulations were employed to offer atomistic insights into the transport mechanism of water molecules across the sub-nanochannels in ML-UiO-66 membranes. In this study, one defect-free and two highly symmetric missing-linker structures were respectively investigated with defect densities of 0, 1/6, and 1/3, corresponding to 0, 4, and 8 linker vacancies per unit cell (denoted also, respectively, as

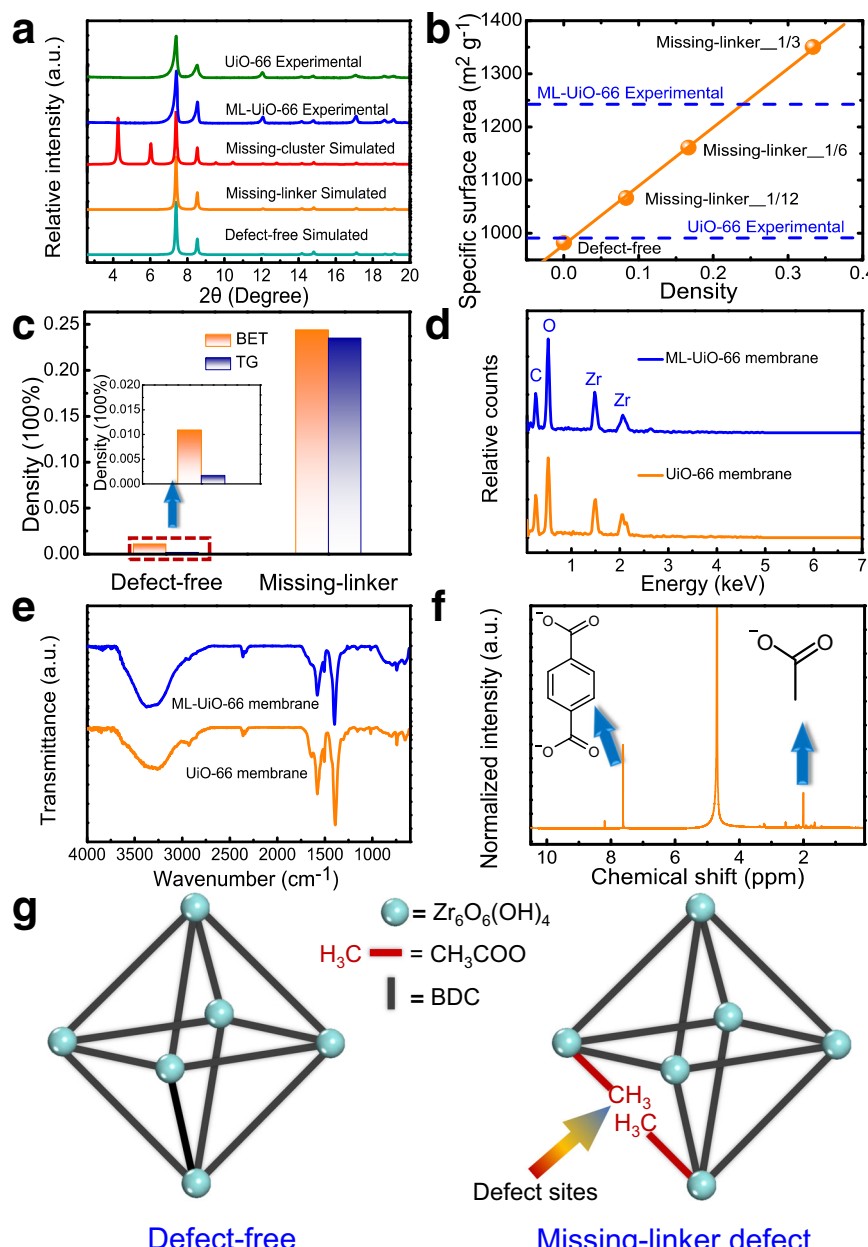

**Fig. 4 Experimental characterization of missing-linker defects in ML-UiO-66 structure. a** Low-angle (2.6–20°, $\lambda = 1.541$ Å) experimental and simulated XRD patterns of UiO-66 structures with different intra-crystalline defects. **b** Simulated and experimental relationship between specific surface area and missing-linker defect density. **c** Defect density comparison between UiO-66 and ML-UiO-66 powders determined by Brunner−Emmet−Teller (BET) and thermogravimetric (TG) results (Supplementary Figs. 24, 25). **d** EDS spectra of UiO-66 and ML-UiO-66 membrane surface. **e** Attenuated total reflection-Fourier transform infrared spectroscopy (ATR-FTIR) spectra of UiO-66 and ML-UiO-66 membrane. **f** Liquid ¹H Nuclear magnetic resonance (NMR) spectrum of ML-UiO-66 powder. **g** Structural representations of UiO-66 (defect-free) and ML-UiO-66 (missing-linker defect).

defect-free, missing-linker_1/6 and missing-linker_1/3; Supplementary Fig. 27). In agreement with experimental observations, our calculations predicted that the presence of missing-linker defects improved the water flux of the membranes without sacrificing their selectivity via beneficially altering the hydrophilicity (i.e., adsorption affinity) and size of confined subnanometer channels (Fig. 5a, b). The introduction of missing-linker defects significantly promoted the rapidly kinetic adsorption of more water molecules with enhanced binding energy (Fig. 5a, c, e right). After adsorption equilibrium, water uptake in the membranes could be maintained at a constant level with operating time during the PV process (Fig. 5c inset). Specially, with enhanced defect density, more water molecules persisted in

the membranes (Fig. 5c inset, Fig. 5e left), while more water molecules also transported from the feed to the permeate side (Fig. 5d). The water flux of an ideal, defect-free UiO-66 membrane having an experimentally comparable thickness is predicated to be $66.9 \pm 22.4$ L m$^{-2}$ h$^{-1}$. In contrast, the water fluxes for ML-UiO-66 membranes are greatly enhanced by as much as 3.6 and 6.1-fold (i.e., $239.4 \pm 43.7$ and $407.7 \pm 26.5$ L m$^{-2}$ h$^{-1}$), for defect densities of 1/6 and 1/3, respectively (Fig. 5f). We should note that the difference between the simulated and experimental water fluxes may be attributed to the spatial arrangements of defects in the experimental sample, which may not be well aligned along the permeation direction in ML-UiO-66, unlike the ideal (aligned) structures studied in simulations.

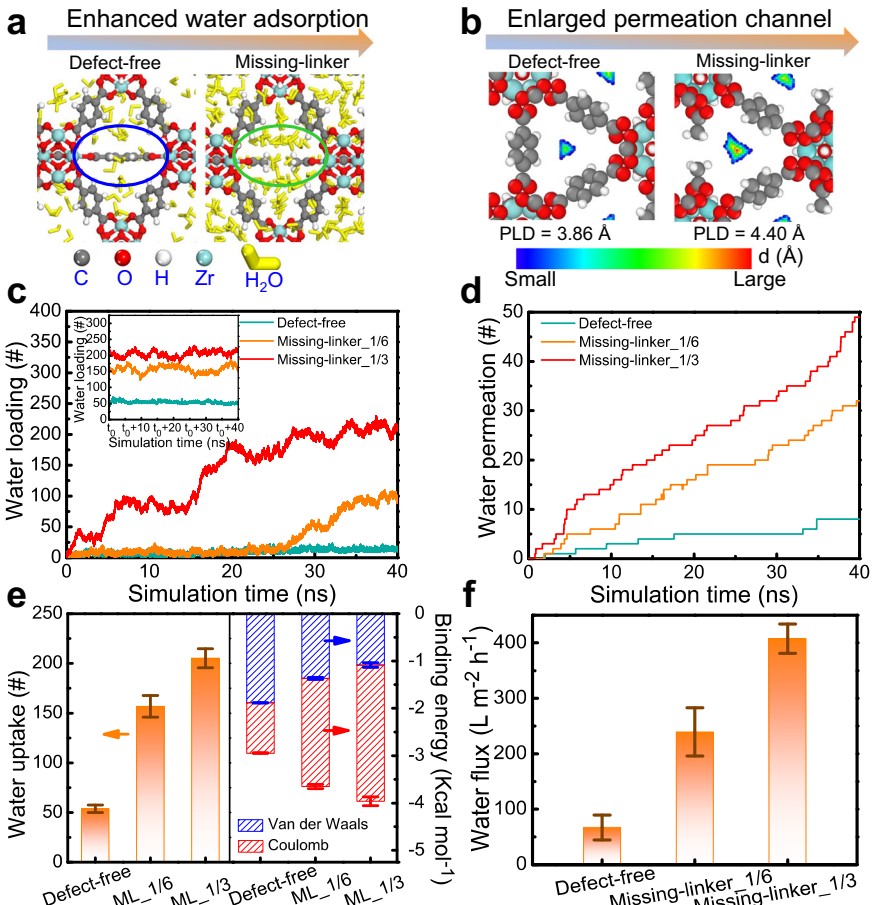

**Fig. 5 Molecular dynamics simulations of water molecule transport through UiO-66 or ML-UiO-66 membranes during a PV process. a** Schematic diagram of defect-enhanced adsorption of water molecules in UiO-66 and ML-UiO-66 structure. **b** Two-dimensional schematic diagram of different sub-nanometer channels for water molecule transport in UiO-66 and ML-UiO-66 structure using pore limiting diameter (PLD) as an evaluation indicator (Note: warmer colors toward red indicates a larger distance. See Supplementary Fig. 29 for more details.) **c** Dynamic adsorption process (i.e., simulated adsorption isotherm curves) of water molecules in UiO-66 and ML-UiO-66 membranes as a function of simulation time (the inset indicates the number of water molecules in UiO-66 and ML-UiO-66 membranes as a function of simulation time after adsorption equilibrium i.e., denoted as $t_0$). **d** Number of permeated water molecules across UiO-66 and ML-UiO-66 membranes as a function of simulation time. **e Left**: Number of water molecules residing in UiO-66 and ML-UiO-66 membranes after adsorption equilibrium (ML: missing linker); **Right**: Binding energy (Van der Waals and Coulombic contributions) between water molecule and the UiO-66/ML-UiO-66 structure. A more negative binding energy value represents a stronger attractive interaction. Details of the binding energy calculations can be seen in the Supplementary Information. **f** Simulated water flux of UiO-66 and ML-UiO-66 membranes. (Numbers that follows the ±signs are standard deviation (SD) in this study.).

Aside from the conventional molecular sieving mechanism, the PV separation process also involves the more dominant adsorption–diffusion mechanism with membrane-to-water adsorption affinity (i.e., adsorption ability) and channel nature (i.e., diffusion ability) as key driving factors, which are highly dependent on the chemical nature and structure of MOF membranes[54]. Compared with UiO-66 membranes, the enhanced flux of ML-UiO-66 membranes can be largely attributed to their enhanced water affinity (between water molecules and defect sites) with enhanced binding energy (Fig. 5a, c, e) and enlarged pore window (connected between central octahedral cage (11 Å) and tetrahedral angle cage (8 Å)) (Fig. 5b, d) due to the presence of missing-linker defects, allowing a more rapid transport of water molecules (2.76 Å) across 3D sub-nanometer channels of ML-UiO-66 membranes. This effect is particularly pronounced for the observed flux difference between defect-free and defective structures (Fig. 5f)[55]. A condensed water phase cannot be formed in a defect-free membrane, due to a lower binding energy between water molecule and defect-free membrane framework (Fig. 5c, e and

Supplementary Fig. 28). By contrast, a continuous water phase can form in membranes with defects, resulting in much faster water transport. Structure comparisons (defect-free, missing-linker_1/3 and missing-linker_1/6, Supplementary Fig. 29) indicate that increasing defect density enlarges the pore window along with increasing the PLD (from 3.86, 4.30–4.40 Å), thus promoting water permeability while rejecting hydrated $Na^+$ ions (7.16 Å) with high selectivity via a size-sieving mechanism. Thus, we can conclude that the introduction of missing-linker defects increases water-to-membrane adsorption affinity (i.e., water adsorption ability, Fig. 5a, e), which is reflected by both enhanced membrane surface hydrophilicity (Supplementary Fig. 21) and water uptake behavior (Supplementary Fig. 22)[55]. Moreover, it also endows faster diffusion transport of more water molecules via enlarging the pore window with a lower diffusion energy barrier of water molecules (Supplementary Fig. 30) across sub-nanometer channels in the ML-UiO-66 membranes. For all these membranes, salt ions were also found to be fully rejected (i.e., 100% rejection, simulation result) by a size-sieving mechanism. Overall, missing-linker defects in ML-UiO-66

membranes are fully demonstrated to beneficially contribute to its promising desalination performance.

## Discussion

An intra-crystalline defect conceptual strategy (i.e., missing-linker) is proposed to rationally design robust ultrathin ML-UiO-66 membranes at a molecular level to enhance water permeation via a PV desalination process. The purposeful introduction of linker defects leads to defect-enhanced permeation, which was confirmed by both experimental and simulation protocols. The introduction of a nanoporous γ-Al$_2$O$_3$ interlayer on scalable coarse ceramic substrates provided more heterogeneous nucleation sites and substantially lowered roughness, favoring the growth of high-quality ultrathin ML-UiO-66 membranes (103 ± 14 nm) with well inter-grown crystals. For ultrathin ML-UiO-66 membranes, besides almost complete salt rejection, high water fluxes (29.8 L m$^{-2}$ h$^{-1}$) were achieved, far outperforming other state-of-the-art zeolite and MOF membranes. Moreover, excellent performance stability in flux and rejection was experimentally confirmed, even for the treatment of hypersaline waters under harsher environments, such as long-term (~10 days) high-temperature (70 °C) chlorine-bearing PV operation. For the ML-UiO-66 structure, defect-compensating ligand was experimentally confirmed to be monocarboxylate group for missing-linker defects, the density of which had an enhanced effect by increasing specific surface area from 990.4 to 1249.0 m$^2$ g$^{-1}$, increasing pore size from 0.508 to 0.568 nm and structural hydrophilicity of three-dimensional sub-nanometer channels enabling fast water transport. Via these molecular-level mechanisms, enhancing missing-linker defects in UiO-66 membranes substantially improved water flux by an enhanced adsorption–diffusion mechanism, while maintaining almost complete salt rejection via a size-sieving mechanism. The design protocols in this study contribute to a concept in utilizing beneficial intra-crystalline defects in MOF membranes as well as provide a more facile strategy in enhancing water permeation performance in desalination or other applications, thus paving a promising way toward molecular-level designable production of high-performance next-generation MOF membranes for more water treatment applications.

## Methods

**Raw materials and chemical reagents**. Raw materials and chemical reagents can be found in the Supplementary Information (Supplementary Note 1, S1.1).

**Fabrication of γ-Al$_2$O$_3$ interlayer**. To address the key issue of poor missing-linker UiO-66 (ML-UiO-66) membrane quality deposited on coarse ZrO$_2$ substrates, dip-coating technique followed by calcination (750 °C for 2 h in air) was then applied to introduce a γ-Al$_2$O$_3$ interlayer onto the ZrO$_2$ substrate using boehmite sol (denoted as ZrO$_2$@γ-Al$_2$O$_3$ substrate)[14,56]. The key role of this interlayer is to provide more heterogeneous nucleation sites (surface –OH groups) and a lower surface roughness for better growth of inter-grown ultrathin ML-UiO-66 membranes.

To fabricate the nanoporous γ-Al$_2$O$_3$ interlayers on ZrO$_2$ substrates, a modified procedure was used to first prepare boehmite sol[56]. In a typical experiment, 15.6306 g aluminum isopropoxide was slowly added into 135 mL stirred ultrapure water to form a solution, which was then refluxed at 84 °C for 1.5 h. Then, 10 mL HNO$_3$ solution (1.8 mol L$^{-1}$) was added as a peptizing agent, and the resulting mixture was then aged at 90 °C for 24 h under reflux. Finally, 13 mL polyvinyl alcohol (PVA) solution (~5 wt.%) was added and then stirred for 2 h under reflux at 90 °C. The ZrO$_2$ substrates with both ends sealed with thread seal tape were dipped into the boehmite sol, and then drawn out at a speed of 5 cm s$^{-1}$. Afterwards, the coated substrates were dried inside a laboratory-made constant-humidity chamber at room temperature (25 °C) for 48 h and then sintered at 750 °C for 2 h in a temperature-programmable furnace at a low heating rate of 1 °C min$^{-1}$. After being naturally cooled to room temperature (25 °C), ZrO$_2$@γ-Al$_2$O$_3$ substrates were obtained (Supplementary Figs. 1, 3, and 4).

**Growth of ultrathin membrane**. ML-UiO-66 membranes were fabricated on coarse ceramic substrates with and without γ-Al$_2$O$_3$ interlayer (i.e., ZrO$_2$ substrates

and ZrO$_2$@γ-Al$_2$O$_3$ substrates) by a one-step in situ solvothermal method (Fig. 1)[39]. A clear and homogenous mother solution was prepared by mixing ZrCl$_4$, H$_2$BDC, CH$_3$COOH, and DMF (with molar ratios of 1:1:X:500, $X$ = 1, 15, 25, 50, 75, respectively) under magnetic stirring for 0.5 h. ZrO$_2$ and ZrO$_2$@γ-Al$_2$O$_3$ substrates were placed vertically into an autoclave filled with the mother solution. The autoclave was placed in an oven and heated at 120 °C for 48 h for membrane growth. After cooling to room temperature (25 °C), the as-synthesized ML-UiO-66 membranes were repeatedly washed with DMF and C$_2$H$_5$OH and then heated at 150 °C for 12 h under vacuum (~60 Pa) for activation. To experimentally reveal the effect of missing-linker defect on desalination performance, UiO-66 (defect-free) membranes were also prepared using the same procedure but without CH$_3$COOH modulator.

**Pervaporation desalination experiments**. Different-salinity feed water sources (5–95 g L$^{-1}$) were first prepared by dissolving NaCl (AR, Tianjian Damao, China) in ultrapure water. The desalination performance and long-term stability of UiO-66 and ML-UiO-66 membranes were then evaluated by a typical pervaporation (PV) process at different operating temperatures (30–90 °C) and feed salt concentrations (5–95 g L$^{-1}$). For PV experiments, the membranes were sealed inside a laboratory-made membrane module, which was heated with feed solution in a thermostatic water bath. The permeate side of the membrane was evacuated using a vacuum pump (~60 Pa). Two cold traps cooled with liquid nitrogen were used to collect permeate (i.e., water) for weighing at constant time intervals after mass transfer equilibrium. The salt concentrations of both feed and permeate sides were analyzed using a conductivity meter (DDSJ-308F, INESA, China). Water flux ($F$, L m$^{-2}$ h$^{-1}$) and salt rejection ($R$, %) were calculated using Eq. (1–2):

$$F = \frac{\Delta m}{A \rho \, t} \times 1000 \quad (1)$$

$$R = 1 - \frac{C_P}{C_F} \quad (2)$$

where $\Delta m$ (g) is the mass of the permeate, $t$ (h) is the sampling interval time, $A$ (m$^2$) is the effective area of the membrane, $\rho$ (g cm$^{-3}$) is the density of water (0.9971 g cm$^{-3}$ at 25 °C), $C_F$ and $C_P$ (g L$^{-1}$) are the salt concentrations of the feed and permeate, respectively.

**Computational simulation details**. Molecular dynamics simulations, implemented in the LAMMPS package[57], were employed to investigate the PV desalination performance and mechanism of both UiO-66 and ML-UiO-66 membranes. The PV simulation system is described with details in Supplementary Fig. 2.

PV simulations were carried out in a canonical (i.e., NVT) ensemble at a temperature of 343 K with a time-step of 1 fs. The system temperature was modulated using the Nosé–Hoover thermostat with a damping factor of 100 time steps[58]. We note that, prior to the PV simulations, each framework structure was first saturated with water molecules via a reverse osmosis process at an applied pressure of ~300 Pa using a simulation set similar to a prior study[32]. It is also worth mentioning that, before sampling PV desalination performance, a sufficiently long simulation was first conducted to reach a steady-state flow. In these calculations, all of the studied membranes, as well as the piston and adsorbing plate, are assumed to be rigid.

To assess the water affinity of UiO-66/ML-UiO-66, molecular dynamics simulations in the NVT ensemble at a temperature of 343 K were also performed for 40 ns to probe the water intruding behavior of an initially empty UiO-66/ML-UiO-66 slab (i.e., to mimic the water intrusion behavior). The number of water molecules inside the framework structure was tracked as a function of the simulation time (Fig. 5c). In addition, the interaction energy of water in framework structure (i.e., between water and framework) were probed using Monte Carlo (MC) simulations in the NVT ensemble implemented in the RASPA package[59]. A single water molecule was inserted in the simulation box (i.e., framework structure). The ensemble-average interaction energy between water and framework as well as its detailed decomposition of van der Waals and Coulombic interactions were quantified (Fig. 5e). We note that, in these MC calculations, the Coulombic interactions were calculated using the Ewald summation technique with a relative error of 10$^{-6}$.

**Characterization**. Detailed characterization can be found in the Supplementary Information (Supplementary Note 1, S1.4).

## Data availability

The authors declare that all data supporting the findings of this study are available within the paper and its supplementary information files or available from the corresponding author upon request.

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

## Acknowledgements

This work was financially supported by the National Natural Science Foundation of China (No. 21876020 and No. 52070033), the National Key Research and Development Project (No. 2019YFA0705803), Youth Top-Notch Talent Program of Talent Project of Revitalizing Liaoning (No. XLYC1807250), the 111 Program of Introducing Talents of Discipline to Universities (No. B13012). The authors gratefully acknowledge the Ohio Supercomputer Center (OSC, USA) for providing computational resources. We also appreciate Prof. Tonghua Wang, Prof. Shouhai Zhang and Prof. Jianhua Yang at DUT (Dalian University of Technology, China), Dr. Shiqiang Wang at UL (University of Limerick, Ireland) for their kind assistance and suggestions.

## Author contributions

Y.D. served both as PI and advisor for this project. M.G. and X.Q. served as co-advisor. Y.D. designed the whole research. X.W. designed, did all the experiments and performed data analysis. X.W. wrote the first draft of the manuscript. Y.D. and M.G. significantly revised the manuscript. K.S., T.T., C.T., and F.Y. gave some suggestions and revised this manuscript. L.L. and Q.L. led the computational part of this work, and Q.L performed all computational simulations. Both L.L and Q.L. contributed to the partial writing of the manuscript.

## Competing interests

The authors declare no competing interests.
