## [Peer Review File · Nature Communications]

Robust Ultrathin Nanoporous MOF Membrane with Intracrystalline Defects for Fast Water TransportREVIEWER COMMENTS

Reviewer #1 (Remarks to the Author):

The following observations are related to the manuscript "Robust ultrathin Nanoporous MOF Membranes with Intra-crystalline Defects for Fast Water Transport"

- It is noted a wrong use of parenthesis: some experimental values included in the manuscript are reported in parenthesis. For example, at page 11 the following sentence can be found: the water flux of ML-UiO-66 membrane increased from (9.1 ± 0.17) to (45.1 ± 1.6) L m⁻² h⁻¹.
- The subheading denominated Introduction is missing
- At page 16, it is not clear how, from the use of BET, TG and XRD allows to conclude that the defects detected were the missing-linker defects. At this regard, since the entire manuscript takes into account the advantages associated to the missing-linker defects, one could also include a brief definition of these morphologic mistakes.
- It is not clear why in the document denominated "supplementary information" mechanical properties of the tested membranes and information on porosity is missing
- A reference corroborating the concepts presented in the following sentence (pag. 14) should be included: This implies that a ML-UiO-66 membrane may be expected to maintain serviceable water flux and salt rejection for as long as 6.8-13.7 years if a typical dosage of NaClO ~ 2-4 ppm is applied
- In the conclusions some remarkable experimental results, especially flux values of the tested membranes, should also be included

On the basis of the abovementioned points, major revision is needed.

Reviewer #2 (Remarks to the Author):

The authors proposed an approach to enhance the water permeability of a pervaporation (PV) membrane via a defect engineering strategy. Although the membrane performance seems impressive, this manuscript cannot be published in Nature Communications at the current stage due to the following reasons.

1. PV is an advanced and relatively energy-intensive technique to dehydrate organic solvents and separate various azeotropes such as p-xylene and its isomers. The usage of PV for desalination can reach 100% salt rejection in theory since salts cannot be vaporized. I am wondering if the authors can further increase the usage of acetic acid, which will not only lead to increased intra-crystalline defects but also inter-crystalline defects. The enhanced water permeability may due to the enhanced inter-crystalline defects. However, the salt rejection rate can still remain 99.9% since salts cannot be vaporized. If the membrane can prevent water intrusion, the salt selectivity will not be compromised by the increased inter-crystalline defects. Thus, to overcome this issue, the authors are suggested to directly try reverse osmosis (RO) for desalination or alcohol dehydration. If the resulting membranes have inter-crystalline defects, ions or alcohols can pass through the membrane. Please refer to these studies: *Angew.Chem. Int.Ed.* 2021, 60,1323 –1331; *J. Am. Chem. Soc.* 2015, 137, 6999–7002.

2. The authors proved that the ML-UiO66 crystals have intra-crystalline defects, but they didn't provide direct evidence that the resulting membranes have enhanced porosity or enlarged pore windows. The pore size distribution of the membranes can be characterized by various techniques, including Positron Annihilation Lifetime Spectroscopy (PALS) or PEG rejection experiments. Please refer to these studies: 10.1002/adma.202006993; *Chemical Engineering Journal* 422 (2021) 130015.

3. The coating of nano-porous γ -Al₂O₃ onto a coarse ceramic substrate is routine and does not deserve highlighting. In fact, commercial ceramic membranes already have such coatings, whether they are γ -Al₂O₃ or α -Al₂O₃ coatings.

4. The authors indicated that the enhanced water permeability of their membranes was due to the enhanced water adsorption affinity and the increased water diffusivity, and the transport process was governed by the solution-diffusion model. However, they only presented simulation data to support, which is not enough. Here are some further suggestions.

- a. Model the process with the solution-diffusion model ($P_i = D_i \times S_i$). Please directly calculate the diffusivity coefficient and solubility coefficient of water molecules through experimental data.
- b. Please try the water adsorption test and calculate the isosteric heat of water adsorption, which can indicate the adsorption affinity of water to UiO and ML-UiO.
- c. Please refer to this study: *Angew. Chem.* 2021, 133, 13191 – 13198

5. Some studies have indicated that an obvious broad diffraction in the 2θ range between 3° and 7° can be observed in defective UiO-66, originating from the cluster defects (*J. Am. Chem. Soc.* 2020, 142, 3174–3183). Why can a similar phenomenon not be observed in this study?

6. The authors compared their results with RO membranes, which is not fair. Please directly compare with PV or MD membranes. The performance data are moderate as compared to (1) Graphene oxide membrane for PV desalination (*J. Mater. Chem. A*, 2019, 7, 18642–18652), in which the flux is $124 \text{ kg m}^{-2} \text{ h}^{-1}$; (2) PVDF membrane for MD desalination (*Journal of Membrane Science* 539 (2017) 34–42), in which the flux is $61.9 \text{ kg m}^{-2} \text{ h}^{-1}$.

Reviewer #3 (Remarks to the Author):

Guiver and coworkers present a very exciting family of ultra-thin MOF membranes that exhibit exquisite water flux alongside requisite stability. They are of interest and may be suitable for publication in *Nature Materials* should the issues raised below be able to be adequately addressed:

1. P6, Line 103-104. It mentioned, "Acetic acid (CH_3COOH) was used as a modulator to rationally quantitatively create missing-linker defects in ML-UiO-66 membranes". The details of how Acetic acid can quantitatively modulate the defects should be included. Is it (the CH_3COOH , as a modulator) a general method that can work on other water-stable MOFs to form a missing-linker defects membrane?

2. Figure 3 f and 3h, 10 days stability was demonstrated in these figures. Wondering how long the membrane performance can hold before lose its stability?

3. Figure 3g, comparisons with other MOFs/materials are made in both RO and PV scenarios. Curious about the experimental performance comparison between the control sample (without CH_3COOH modulator) and ML-UiO-66 using the same membrane fabrication method. Same for the performance in Figure 3 f and Figure 3h (comparison between the control sample and ML-UiO-66 in terms of stability in harsh conditions and salt rejection performance?). The comparison between the control and ML-UiO-66 in the Molecular Dynamics Simulations part is very clear.

4. Simple explanation about Figure 4a will help the readers' understanding rather than just mention this figure. For example P16, Line 293-294: "In addition, missing-cluster defects can be excluded according to low-angle experimental and simulated XRD patterns (Fig. 4a)".

5. P16, Line 303-305: confused by the description. $\sim 0.17\%$ and $\sim 1.09\%$ are supposed for the defect-free samples rather than the missing-linkers UiO-66 density? But, missing-linker defect densities were also included here?

6. More details for terminal group confirmation for ML-UiO-66 in P17 lines 312-315? Comparison between defect-free and ML-UiO-66 in EDS (figure 4d) and FTIR (figure 4e); For figure 4e, is it supposed to be the comparison between the powder and membrane

ML-UIO-66 rather than defect-free versus ML-UIO-66?

7. Maybe experimental Contact angle performance can be the further supplementary evidence for the hydrophilicity (water adsorption affinity) of ML-UIO-66 which was demonstrated in the Molecular Dynamics Simulations (Figure 5)?

Responses to Referees' Comments

Responses to Reviewer-1's comments

General Comments: The following observations are related to the manuscript "Robust Ultrathin Nanoporous MOF Membranes with Intra-crystalline Defects for Fast Water Transport". On the basis of the following points, major revision is needed.

Response: We appreciate the reviewer for kindly providing the helpful comments. We have carefully considered all the comments/suggestions, given point-by-point responses (shown as follows), done additional experiments, made new/revised figures and tables, and made revisions to improve the quality of our manuscript and supplementary material.

R1-1. It is noted a wrong use of parenthesis: some experimental values included in the manuscript are reported in parenthesis. For example, at page 11 the following sentence can be found: the water flux of ML-UiO-66 membrane increased from (9.1 ± 0.17) to (45.1 ± 1.6) $\text{L m}^{-2} \text{h}^{-1}$.

Response: Thank you for this comment. To address the reviewer's concern, we have carefully checked all the parentheses throughout the manuscript and deleted all the wrongly used parentheses (Line 198–202, Line 205–207, Line 211–214, Page 10–11, revised manuscript).

The revisions are also shown as follows:

Line 198–202, Page 10, revised manuscript:

With increasing operation temperature from 30 to 90°C, the water flux of ML-UiO-66 membrane increased from 9.1 ± 0.7 to 45.1 ± 1.6 $\text{L m}^{-2} \text{h}^{-1}$ due to an enhanced driving force for faster water molecular diffusion transport within the 3D sub-nanometer channels, while maintaining stable salt rejection of > 99.8% (Fig. 3d).

Line 205–207, Page 10, revised manuscript:

Even at 30°C (close to room temperature), a high water flux of 9.14 ± 0.70 $\text{L m}^{-2} \text{h}^{-1}$ was achieved with high rejection (~99.9%) for 35 g L^{-1} NaCl solution.

Line 211–214, Page 11, revised manuscript:

Nevertheless, salt rejection was maintained at a consistently stable level ($> 99.8\%$) even for ultrahigh salinity feed (95 g L^{-1}) (Fig. 3e), while water flux still remained a high level ($16.3 \pm 0.3 \text{ L m}^{-2} \text{ h}^{-1}$), indicating its promising potential for hypersaline water treatment.

R1-2. The subheading denominated Introduction is missing.

Response: According to the Author Guideline of Nature Communications, we have added the subsection heading “Introduction” in the revised manuscript. Thank you very much!

R1-3. At page 16, it is not clear how, from the use of BET, TG and XRD allows to conclude that the defects detected were the missing-linker defects. At this regard, since the entire manuscript takes into account the advantages associated to the missing-linker defects, one could also include a brief definition of these morphologic mistakes.

Response: We appreciate the reviewer for the valuable suggestion. It is reasonable to conclude the presence of missing-linker defects based on the results of BET, TG and XRD, which has been also well documented in previous reports.³⁷ Please see more details about the discussion of BET and TG results (Page S44–S47, revised supplementary material). To clarify this issue, we have also made revisions in the revised manuscript (Line 291–303, Page 15), which are as follows:

Compared to defect-free UiO-66, the existence of certain defects in the ML-UiO-66 structure is confirmed by increased specific surface area and lower linker weight loss based on BET and TG analysis (Figs. S24 and S25, Table S10). Generally, there are two types of intra-crystalline defects, which are missing-linker (Fig. 4g) and missing-cluster, in defective UiO-66 MOF structures. Compared with defect-free UiO-66 structure, a new broad XRD diffraction reflection is present in the 2θ range of $3 - 7^\circ$ only for missing-cluster defective UiO-66 structure due to the presence of *reo* phase.⁵⁵

In contrast, there is no such a characteristic diffraction reflection for both missing-linker defective and defect-free UiO-66 structures. Therefore, in our work, missing-cluster defects can be excluded according to the low-angle experimental and simulated XRD patterns in which there is no broad diffraction reflection in the 2θ range of $3 - 7^\circ$ (Fig. 4a).^{56,57} In summary, we can reasonably infer that the defects in ML-UiO-66 membrane should be missing-linker defects.

To address the reviewer's concern, we have given a brief definition of the morphological structure of missing-linker defects in UiO-66 MOF (Line 303–306, Page 15–16, revised manuscript), which is as follows:

In a defect-free UiO-66 structure, each BDC²⁻ linker coordinates with two Zr₆ clusters. In contrast, for a missing-linker defect structure, the absence of one BDC linker with negative charge causes the formation of two defect centers having four coordinated unsaturated Zr sites.³⁷

Reference:

55. Feng, X.; Hajek, J.; Jena, H. S.; Wang, G.; Veerapandian, S. K. P.; Morent, R.; De Geyter, N.; Leyssens, K.; Hoffman, A. E. J.; Meynen, V.; Marquez, C.; De Vos, D. E.; Van Speybroeck, V.; Leus, K.; Van Der Voort, P., Engineering a highly defective stable UiO-66 with tunable lewis- bronsted acidity: the role of the hemilabile linker. *J. Am. Chem. Soc.* **2020**, *142* (6), 3174-3183.

56. Cliffe, M. J.; Wan, W.; Zou, X.; Chater, P. A.; Kleppe, A. K.; Tucker, M. G.; Wilhelm, H.; Funnell, N. P.; Coudert, F. X.; Goodwin, A. L., Correlated defect nanoregions in a metal-organic framework. *Nat. Commun.* **2014**, *5*, 4176.

57. Liu, L.; Chen, Z.; Wang, J.; Zhang, D.; Zhu, Y.; Ling, S.; Huang, K. W.; Belmabkhout, Y.; Adil, K.; Zhang, Y.; Slater, B.; Eddaoudi, M.; Han, Y., Imaging defects and their evolution in a metal-organic framework at sub-unit-cell resolution. *Nat. Chem.* **2019**, *11* (7), 622-628.

37. Feng, Y.; Chen, Q.; Jiang, M.; Yao, J., Tailoring the properties of UiO-66 through defect engineering: A review. *Ind. Eng. Chem. Res.* **2019**, *58* (38), 17646-17659.

R1-4. It is not clear why in the document denominated “supplementary information” mechanical properties of the tested membranes and information on porosity is missing.

Response: Thank you for pointing this omission out. To address the reviewer's concern, we have done additional experiments and simulations (Page S10, revised supplementary material), and included new discussion, and figures about mechanical properties and porosity (Fig. S9, Page S21, revised supplementary material; Fig. S14, Page S28, revised supplementary material), which are shown as follows:

Page S10, revised supplementary material:

The mechanical strength of the ZrO₂ substrate was measured using the three-point bending method by a universal testing machine (AGS-10KN, Shimadzu Ltd., Japan). The bulk porosity of ZrO₂ substrate and γ -Al₂O₃ interlayer was measured by the gravimetric method while surface porosity was calculated by Image J software based on their surface FE-SEM images. The porosity of UiO-66 and ML-UiO-66 membranes was simulated by zeo++ software.

Fig. S9, Page S21, revised supplementary material:

The flexural strength of membrane substrate is 24.96 ± 1.81 MPa is almost equal to that of the tested membranes, since the strength of ultrathin UiO-66 layer can be considered as negligible.

Fig. S9. The load-deflection relationship of membrane substrate.

Fig. S14, Page S28, revised supplementary material:

Fig. S14. The bulk porosity and surface porosity of ZrO₂ substrate and γ-Al₂O₃ interlayer (a) and simulated porosities for UiO-66 and ML-UiO-66 membranes using zeo++ software (b).

R1-5. A reference corroborating the concepts presented in the following sentence (page. 14) should be included: This implies that a ML-UiO-66 membrane may be expected to maintain serviceable water flux and salt rejection for as long as 6.8-13.7 years if a typical dosage of NaClO ~ 2-4 ppm is applied.

Response: We appreciate the reviewer's good suggestion. To address the reviewer's concern, we have added two references (Reference 6, Reference 54) in the revised manuscript (Line 271, Page 14, revised manuscript), which are also shown as follows:

6. Yao, Y.; Zhang, P.; Jiang, C.; DuChanois, R. M.; Zhang, X.; Elimelech, M., High performance polyester reverse osmosis desalination membrane with chlorine resistance. *Nat. Sustain.* **2021**, *4*, 138-146.

54. Wang, Y. H.; Wu, Y. H.; Tong, X.; Yu, T.; Peng, L.; Bai, Y.; Zhao, X. H.; Huo, Z. Y.; Ikuno, N.; Hu, H. Y., Chlorine disinfection significantly aggravated the biofouling of reverse osmosis membrane used for municipal wastewater reclamation. *Water Res.* **2019**, *154*, 246-257.

R1-6. In the conclusions some remarkable experimental results, especially flux values of the tested membranes, should also be included.

Response: Thank you for this very helpful suggestion. To highlight our work, we have further added some experimental results (such as membrane thickness and water flux) in the Conclusions (Line 423–437, Page 21–22, revised manuscript), which are shown as follows:

The introduction of a nanoporous γ -Al₂O₃ interlayer on scalable coarse ceramic substrates provided more heterogeneous nucleation sites and substantially lowered roughness, favoring the growth of high-quality ultrathin ML-UiO-66 membranes (103 ± 14 nm) with well inter-grown crystals. For ultrathin ML-UiO-66 membranes, besides almost complete salt rejection, high water fluxes (29.8 L m⁻² h⁻¹) were achieved, far outperforming other state-of-the-art zeolite and MOF membranes. Moreover, excellent performance stability in flux and rejection was experimentally confirmed, even for the treatment of hypersaline waters under harsher environments, such as long-term (~10 days) high-temperature (70°C) chlorine-bearing PV operation. For the ML-UiO-66 structure, defect-compensating ligand was experimentally confirmed to be monocarboxylate group for missing-linker defects, the density of which had an enhanced effect by increasing specific surface area from 990.4 to 1249.0 m² g⁻¹, increasing pore size from 0.508 to 0.568 nm and structural hydrophilicity of three dimensional sub-nanometer channels enabling fast water transport.

Response to Reviewer-2's comments

General Comments: The authors proposed an approach to enhance the water permeability of a pervaporation (PV) membrane via a defect engineering strategy. Although the membrane performance seems impressive, this manuscript cannot be published in Nature Communications at the current stage due to the following reasons.

Response: We appreciate the reviewer for kindly providing some positive comments on the significance of our work as well as some useful references. We have carefully considered all the comments/suggestions, done additional experiments, added new/revised figures and tables (Fig. 3g, Fig. 4d, Fig. 4e, Table S2, Fig. S9, Figs. S14–S15, Table S4–S5, Figs. S18–S20, Fig. S22 and Table S7), and made revisions to clarify the significance of our work and to improve the quality of our manuscript and supplementary material.

R2-1. PV is an advanced and relatively energy-intensive technique to dehydrate organic solvents and separate various azeotropes such as p-xylene and its isomers. The usage of PV for desalination can reach 100% salt rejection in theory since salts cannot be vaporized. I am wondering if the authors can further increase the usage of acetic acid, which will not only lead to increased intra-crystalline defects but also inter-crystalline defects. The enhanced water permeability may be due to the enhanced inter-crystalline defects. However, the salt rejection rate can still remain 99.9% since salts cannot be vaporized. If the membrane can prevent water intrusion, the salt selectivity will not be compromised by the increased inter-crystalline defects. Thus, to overcome this issue, the authors are suggested to directly try reverse osmosis (RO) for desalination or alcohol dehydration. If the resulting membranes have inter-crystalline defects, ions or alcohols can pass through the membrane. Please refer to these studies: *Angew. Chem. Int. Ed.* 2021, 60, 1323–1331; *J. Am. Chem. Soc.* 2015, 137, 6999–7002.

Response: Thank you for providing valuable comments and references. We also agree with the reviewer's opinion about the effects of defects on desalination performance. As shown in Fig. S11 (Page S24, revised supplementary material), the amount of acetic

acid played an important role in not only creating intra-crystalline defects but affecting membrane morphology. The addition of excess CH_3COOH resulted in the formation of isolated ML-UiO-66 crystals (not well-intergrown membrane), which are believed to be unsuitable for PV application. In our work, the optimized ratio of $\text{Zr}^{4+}:\text{CH}_3\text{COOH}$ was 1:25, where well-intergrown ML-UiO-66 membranes were formed with intra-crystalline defects.

To address the reviewer's concern about the integrity of UiO-66 and ML-UiO-66 membranes, we have done additional experiments including single-gas permeation (Table S4, Page S30, revised supplementary material) and RO desalination performance (Table S5, Page S31, revised supplementary material). The ideal selectivities of H_2/CH_4 for UiO-66 and ML-UiO-66 membranes are experimentally determined to be 30.5 and 17.1, both of which are much higher than the Knudsen diffusion coefficient (2.8). This result indicates the membrane integrity has negligible inter-crystalline defects for UiO-66 and ML-UiO-66 membranes.

For RO desalination, as shown in Table S5, the rejection of NaCl for UiO-66 and ML-UiO-66 membranes were 48.6% and 46.5%, respectively. The aperture sizes of UiO-66 and ML-UiO-66 membranes were experimentally determined to be 5.86 and 6.28 Å (Fig. S15, Page 29, revised supplementary material), both of which are between H_2O (2.8 Å) and hydrated Na^+ (7.2 Å). Theoretically, both UiO-66 and ML-UiO-66 membranes are able to reject hydrated Na^+ . However, low rejection of NaCl can be also similarly observed for UiO-66 (47.0%)³³ and UiO-66-(OH) membranes (45.0%)³² in previous reports. Such abnormally low salt rejection can be ascribed to the dehydration or partial dehydration effect during the RO desalination process, which has been well confirmed in polymeric³⁷ and MOF nanochannels.³⁸ The hydrated and fully dehydrated ionic diameters of Na^+ are 7.16 and 1.90 Å, respectively.³⁸ Such a dehydration or partial dehydration effect occurred, where hydrated Na^+ ions should undergo a dehydration or partial dehydration to enter the nanochannels of UiO-66 or ML-UiO-66 membranes under a high pressure driven process, and then rehydrated by water molecules when they exit the membrane pore into aqueous media.^{37, 38}

In addition to the unsatisfactory rejection, the water permeances are also as low as 0.308 and 0.324 L m⁻² h⁻¹ bar⁻¹ for UiO-66 and ML-UiO-66 membranes (Table S5). Moreover, the RO process is inherently unsuitable for the desalination of hypersaline waters, because of their very high osmotic pressure. In contrast, under low transmembrane pressure, the PV process can be used as an alternative promising technology using ML-UiO-66 membranes (high water flux, high rejection and good operating stability), enabling the efficient treatment of hypersaline waters at harsh conditions, which has been well confirmed in our work.

We have now included additional discussion about single-gas permeation and RO desalination in supplementary material, as follows:

Table S4, Page S30, revised supplementary material:

The ideal selectivities of H₂/CH₄ for UiO-66 and ML-UiO-66 membranes are experimentally determined to be 30.5 and 17.1, both of which are much higher than the Knudsen diffusion coefficient (2.8). This indicates the membrane integrity had negligible inter-crystalline defects for UiO-66 and ML-UiO-66 membranes.

Table S4. The single-gas permeation results of UiO-66 and ML-UiO-66 membranes.

Membrane	H ₂ permeance (GPU)	CH ₄ permeance (GPU)	$\alpha_{\text{H}_2/\text{CH}_4}$ (ideal)	$\alpha_{\text{H}_2/\text{CH}_4}$ (Knudson)
UiO-66	1864.1	61.1	30.5	2.8
ML-UiO-66	662.7	38.7	17.1	2.8

Table S5, Page S31, revised supplementary material:

For RO desalination, as shown in Table S5, the rejection of NaCl for UiO-66 and ML-UiO-66 membranes were 48.6% and 46.5%, respectively. The aperture sizes of UiO-66 and ML-UiO-66 membranes were experimentally determined to be 5.86 and 6.28 Å (Fig. S14), both of which are between H₂O (2.8 Å) and hydrated Na⁺ (7.2 Å). Theoretically, both UiO-66 and ML-UiO-66 membranes are able to reject hydrated Na⁺. However, low rejection of NaCl can be also similarly observed for UiO-66 (47.0%)³³ and UiO-66-(OH) membranes (45.0%)³² in previous reports. Such abnormally low salt rejection can be ascribed to the dehydration or partial dehydration effect during the RO

desalination process, which has been well confirmed in polymeric³⁷ and MOF nanochannels.³⁸ The hydrated and fully dehydrated ionic diameter of Na⁺ are 7.16 and 1.90 Å, respectively.³⁸ Such a dehydration or partial dehydration effect occurred, where Na⁺ ions should undergo a dehydration or partial dehydration process to enter the nanochannels of UiO-66 or ML-UiO-66 membranes under a high pressure driven process, and then rehydrated by water molecules when they exit the membrane pore into aqueous media.^{37, 38}

Table S5. Desalination performance of UiO-66 and ML-UiO-66 membranes (0.20 wt.% NaCl was applied as feeds at 20 ± 2°C under a pressure difference of 3.0 bar.).

Membrane	Water permeance (L m ⁻² h ⁻¹ bar ⁻¹)	Rejection (%)
UiO-66	0.308	48.6%
ML-UiO-66	0.324	46.5%

Reference:

32. Wang, X.; Zhai, L.; Wang, Y.; Li, R.; Gu, X.; Yuan, Y. D.; Qian, Y.; Hu, Z.; Zhao, D., Improving water-treatment performance of zirconium metal-organic framework membranes by postsynthetic defect healing. *ACS Appl. Mater. Interfaces* **2017**, *9* (43), 37848-37855.
33. Liu, X.; Demir, N. K.; Wu, Z.; Li, K., Highly water-stable zirconium metal-organic framework UiO-66 membranes supported on alumina hollow fibers for desalination. *J. Am. Chem. Soc.* **2015**, *137* (22), 6999-7002.
37. Lu, C.; Hu, C.; Ritt, C. L.; Hua, X.; Sun, J.; Xia, H.; Liu, Y.; Li, D. W.; Ma, B.; Elimelech, M.; Qu, J., In situ characterization of dehydration during ion transport in polymeric nanochannels. *J. Am. Chem. Soc.* **2021**, *143* (35), 14242-14252.
38. Zhang, H.; Hou, J.; Hu, Y.; Wang, P.; Ou, R.; Jiang, L.; Liu, J. Z.; Freeman, B. D.; Hill, A. J.; Wang, H., Ultrafast selective transport of alkali metal ions in metal organic frameworks with subnanometer pores. *Sci. Adv.* **2018**, *4*, eaaq0066.

R2-2. The authors proved that the ML-UiO-66 crystals have intra-crystalline defects, but they didn't provide direct evidence that the resulting membranes have enhanced porosity or enlarged pore windows. The pore size distribution of the membranes can be

characterized by various techniques, including Positron Annihilation Lifetime Spectroscopy (PALS) or PEG rejection experiments. Please refer to these studies: 10.1002/adma.202006993; Chemical Engineering Journal 422 (2021) 130015.

Response: We appreciate the reviewer for this comment. In our original manuscript (Line 292, Page 16) and supplementary material (Fig. S18, Table S6, Page S33–34), we had indeed provided the direct experimental evidence about the pore volume (0.488 vs. 0.619 cm³ g⁻¹), pore size (0.508 vs. 0.568 nm) and its distribution curves of UiO-66 and ML-UiO-66 based on nitrogen adsorption-desorption experiments.

To address the reviewer's concern, we have now also included solute rejection experiments to determine the pore size distribution of UiO-66 and ML-UiO-66 membranes (Fig. S15, Page S29, revised supplementary material).²⁵ The average pore size was enhanced due to the introduction of intra-crystalline defects (0.586 nm for UiO-66 membrane and 0.628 nm for ML-UiO-66 membrane).

We have now provided the experiment procedures (Page S10–S11, revised supplementary material) and the pore size distribution results of UiO-66 and ML-UiO-66 membranes (Fig. S15, Page S29, revised supplementary material), which are as follows:

Page S10–S11, revised supplementary material:

S1.7 Pore Size Distribution

The mean pore size and pore size distribution of UiO-66 and ML-UiO-66 membranes were experimentally determined by the solute rejection method using a laboratory-made dead-end filtration setup.^{24, 25} A series of ~200 ppm PEG or glycerol solutions (molecular weights: 92, 200, 400 and 600 g mol⁻¹) were used as the feed solution. The operation transmembrane pressure was 3 bar. The concentration of feed (C_f , ppm) and permeate (C_p , ppm) were measured using a total organic carbon analyzer (TOC/TN analyzer, multi N/C 2100S, Germany). The effective solute rejection (R' , %) was calculated by Eq. S3:

$$R' = \left(1 - \frac{C_p}{C_f}\right) \times 100\% \quad (\text{S3})$$

Table S2. Molecular weight and Stokes diameters of the organic solutes.

Solute	Molecular weight M_w (g mol ⁻¹)	d_s (nm)
Glycerol	92	0.52
PEG 200	200	0.64
PEG 400	400	0.94
PEG 600	600	1.18

Based on molecular weights, the dependence of Stokes diameters (d_s , nm) on molecular weight M_w (g mol⁻¹) is shown as Eq. S4:

$$d_s = 33.46 \times 10^{-12} \times M_w^{0.557} \quad (\text{S4})$$

Then, the solute rejection was plotted against Stokes diameter on a log-normal probability diagram and linear regression was performed. The mean effective pore diameter (μ_p , nm) is the size of a solute where its rejection is 50%. The geometric standard deviation (σ_p) of a membrane is the size ratio of solutes with rejections of 84.13% and 50%. The pore size distribution of the membrane was generated using Eq. S5:

$$\frac{dR(d_p)}{d(d_p)} = \frac{1}{d_p \ln \sigma_p \sqrt{2\pi}} \exp \left\{ -\frac{(\ln d_p - \ln \mu_p)^2}{2(\ln \sigma_p)^2} \right\} \quad (\text{S5})$$

where d_p (nm) is the effective pore diameter.

Fig. S15, Page S29, revised supplementary material:

Fig. S15. Pore size distribution of UiO-66 and ML-UiO-66 membranes based on the solute rejection method.

Reference:

24. Ma, D.; Han, G.; Gao, Z. F.; Chen, S. B., Continuous UiO-66-type metal-organic framework thin film on polymeric support for organic solvent nanofiltration. *ACS Appl. Mater. Interfaces* **2019**, *11* (48), 45290-45300.
25. Li, B.; Japip, S.; Lai, J.-Y.; Chung, T.-S., Revitalize integrally skinned hollow fiber membranes with spatially impregnated 3D-macrocycles for organic solvent nanofiltration. *Chem. Eng. J.* **2021**, *422*, 130015.

R2-3. The coating of nano-porous γ -Al₂O₃ onto a coarse ceramic substrate is routine and does not deserve highlighting. In fact, commercial ceramic membranes already have such coatings, whether they are γ -Al₂O₃ or α -Al₂O₃ coatings.

Response: Thank you for pointing this out. We agree with the reviewer's opinion that the coating of nanoporous γ -Al₂O₃ onto a coarse ceramic substrate is routine, since tubular single-channel or multi-channel γ -Al₂O₃ or α -Al₂O₃ ceramic membranes are commercially available. However, different from commercial membranes, γ -Al₂O₃@ZrO₂ membrane developed in our work has a hollow fiber configuration, featuring much smaller dimensions and much higher packing density (Figs. 1 and 2c). The aim of coating γ -Al₂O₃ layer is indeed not our highlight but a necessary step for subsequent fabrication of ML-UiO-66 MOF membranes. To address the reviewer's concern, we have made revisions in section Introduction (Line 96-99, Page 5, revised manuscript), which are as follows:

Ultrathin ML-UiO-66 membranes were fabricated *via* a cost-effective *in situ* liquid phase growth method on nanoporous γ -Al₂O₃ interlayer, which was coated on scalable macroporous hollow fiber ceramic substrate using a routine dip-coating method.

R2-4. The authors indicated that the enhanced water permeability of their membranes was due to the enhanced water adsorption affinity and the increased water diffusivity, and the transport process was governed by the solution-diffusion model. However, they

only presented simulation data to support, which is not enough. Here are some further suggestions.

- a. Model the process with the solution-diffusion model ($P_i = D_i \times S_i$). Please directly calculate the diffusivity coefficient and solubility coefficient of water molecules through experimental data.
- b. Please try the water adsorption test and calculate the isosteric heat of water adsorption, which can indicate the adsorption affinity of water to UiO and ML-UiO.
- c. Please refer to this study: Angew. Chem. 2021, 133, 13191 – 13198

Response: We thank the reviewer for these useful comments and suggestions as well as the useful reference, which are very helpful for us to further clarify the mechanism of water transport and to improve the quality of our work. We have also referred to the suggested reference and other related literature. To address the reviewer's concerns, we have done additional experiments (S1.8 and S1.9, Page S11–S13, revised supplementary material), included a new table (Table S7, Page S39, revised supplementary material) and revised a figure (Fig. S22, Page S38, revised supplementary material).

- a) The methods of determining diffusivity coefficient and adsorption coefficient of water molecules are shown as follows (S1.9, Page S12–S13, revised supplementary material):

S1.9 Measurements of water permeability, adsorption coefficient and diffusivity coefficient²⁷

For the water adsorption coefficient measurements, the UiO-66 and ML-UiO-66 membranes were immersed into DI water for 2 days to fully hydrate the membranes. Then, any water droplets on the membrane surface were gently removed with a dry tissue and the wet membrane (W_{wet} , g) was quickly weighed using an analytical balance. After drying in a vacuum oven at 80°C for 24 h, the dry membrane (W_{dry} , g) was weighed.

The volume fraction of water is related to the water adsorption coefficient, $K_w(-)$.

The $K_w(-)$ is defined as Eq. S8:

$$K_w = \frac{(W_{\text{wet}} - W_{\text{dry}})/\rho_w}{(W_{\text{wet}} - W_{\text{dry}})/\rho_w + V_{\text{dry}}} \quad (\text{S8})$$

where V_{dry} (cm^3) is the volume of the dry membrane, ρ_w (g cm^{-3}) is the density of water.

The hydraulic water permeability (P_w^H , $\text{L } \mu\text{m m}^{-2} \text{h}^{-1} \text{bar}^{-1}$) was measured using DI water as a feed at room temperature (25°C) during PV process. The hydraulic water permeability was calculated by Eq. S9:

$$P_w^H = \frac{\Delta V}{A \Delta t} \frac{l}{\Delta P} \quad (\text{S9})$$

where ΔV (L) is the volume of the permeated water, A (m^2) is the effective membrane areas, Δt (h) is the test time interval, l (μm) is the hydrated membrane thickness, and ΔP (bar) is the transmembrane pressure difference.

Like the solution–diffusion mechanism in polymeric membranes, water molecules transported through UiO-66 and ML-UiO-66 membranes following the adsorption–diffusion mechanism during PV process. The intrinsic hydraulic water permeability (P_w , $\text{cm}^2 \text{s}^{-1}$) of a membrane is associated with P_w^H , which is calculated as Eq. S10:

$$P_w = D_w K_w = P_w^H \frac{RT}{V_w} \quad (\text{S10})$$

where D_w (-) is the water adsorption coefficient, R ($8.314 \text{ J mol}^{-1} \text{ K}^{-1}$) is the ideal gas constant, and T (K) is the absolute temperature during the permeability measurements.

Reference

27. Lee, T. H.; Oh, J. Y.; Jang, J. K.; Moghadam, F.; Roh, J. S.; Yoo, S. Y.; Kim, Y. J.; Choi, T. H.; Lin, H.; Kim, H. W.; Park, H. B., Elucidating the Role of Embedded Metal–Organic Frameworks in Water and Ion Transport Properties in Polymer Nanocomposite Membranes. *Chem. Mater.* **2020**, *32* (23), 10165-10175.

The results of water diffusivity and adsorption coefficient are shown as follows (Table S7, Page S39, revised supplementary material).

Table S7. Adsorption, water diffusivity and permeability of ceramic-based UiO-66 and ML-UiO-66 membranes.

Membrane	Adsorption (-)	Diffusivity ($\times 10^{-6} \text{ cm}^2 \text{ s}^{-1}$)	Permeability ($\times 10^{-6} \text{ cm}^2 \text{ s}^{-1}$) (P=D×S)
UiO-66	0.40	40.22	160.86
ML-UiO-66	0.43	48.49	208.52

b) The experimental procedure of determining the isosteric heat of adsorption were shown as follows (S1.8, Page S11–S12, revised supplementary material):

S1.8 Isosteric heat of adsorption

The isosteric heat of adsorption can be determined by the Clausius-Clapeyron equation as Eq. S6:

$$Q_{st} = -RT^2 \left(\frac{\partial \ln p}{\partial T} \right)_{\theta} \quad (\text{S6})$$

where Q_{st} (kJ mol^{-1}) is the isosteric heat of adsorption, R ($8.314 \text{ J mol}^{-1} \text{ K}^{-1}$) is the gas content, T (K) is the absolute temperature, P (bar) is the pressure, and θ (%) is the sorbed amount. Integration of the Clausius-Clapeyron equation (Eq. S6) gives Eq. S7:

$$\ln p = \frac{Q_{st}}{RT} + C \quad (\text{S7})$$

In this study, adsorption isotherm data measured at 298.15 K and 303.15 K was used to obtain the plots of adsorption heat. The adsorption heat at a given uptake was calculated from the fitted slopes of the equation S7.²⁶

The revised figure as well as additional discussion (Fig. S22, Page S38, revised supplementary material) are as follows:

At a relative humidity of 95% (25°C), the water adsorption capacities of UiO-66 and ML-UiO-66 are 19.82 and 33.1 wt.%, respectively (Fig. S22). Such an enhancement of 67.2% indicates that the introduction of missing-linker defects in the ML-UiO-66 structure enhances its water molecule adsorption ability. Also, at different temperatures, the water adsorption behaviors of UiO-66 and ML-UiO-66 have a similar variation tendency. The initial isosteric heat of water adsorption of ML-UiO-66 is higher than

UiO-66 (51.3 vs. 31.9 kJ mol⁻¹, Fig. S22d), which implies a more favorable interaction between ML-UiO-66 and water molecules than UiO-66. That is to say, ML-UiO-66 has better water adsorption ability than UiO-66.²⁶

Fig. S22. Water uptake behavior of UiO-66 and ML-UiO-66 structures. Water adsorption isotherms of UiO-66 and ML-UiO-66 measured at 25°C (a), 30°C (b) and 35°C (c). (d) Initial isosteric heat of water adsorption (Q_{st}) of UiO-66 and ML-UiO-66 (the inset shows the Q_{st} of UiO-66 and ML-UiO-66 as a function of water uptake capacity).

Reference

26. Lee, T. H.; Jung, J. G.; Kim, Y. J.; Roh, J. S.; Yoon, H. W.; Ghanem, B. S.; Kim, H. W.; Cho, Y. H.; Pinnau, I.; Park, H. B., Defect engineering in metal-organic framework towards advanced mixed matrix membranes for efficient propylene-propane separation. *Angew. Chem. Int. Ed.* **2021**, *60*, 13081-13088.

R2-5. Some studies have indicated that an obvious broad diffraction in the 2θ range between 3 and 7° can be observed in defective UiO-66, originating from the cluster defects (J. Am. Chem. Soc. 2020, 142, 3174–3183). Why can a similar phenomenon not be observed in this study?

Response: Thank you for the good comment, and for kindly providing this useful reference. A different phenomenon in XRD patterns can be ascribed to the presence of the different defect type between our work (i.e., missing-linker defect) and the previous report (i.e., missing-cluster defect) (*J. Am. Chem. Soc.* 2020, 142, 3174–3183). Generally, there are two types of intra-crystalline defects, including missing-linker and missing-cluster, in defective UiO-66 MOF structure. Compared with defect-free UiO-66 structure, a new broad XRD diffraction reflection is present in the 2θ range of $3 - 7^\circ$ only for missing-cluster defective UiO-66 structure due to the presence of *reo* phase.⁵⁵ In contrast, there is no such characteristic diffraction reflection for both missing-linker defective and defect-free UiO-66 structures.

To address the reviewer's concern, we have included revisions (Line 293–303, Page 15, revised manuscript), which are as follows:

Generally, there are two types of intra-crystalline defects, including missing-linker (Fig. 4g) and missing-cluster, in defective UiO-66 MOF structure. Compared with defect-free UiO-66 structure, a new broad XRD diffraction reflection is present in the 2θ range of $3 - 7^\circ$ only for missing-cluster defective UiO-66 structure due to the presence of *reo* phase.⁵⁵ In contrast, there is no such characteristic diffraction reflection for both missing-linker defective and defect-free UiO-66 structures. Therefore, in our work, missing-cluster defects can be excluded according to the low-angle experimental and simulated XRD patterns in which there is no broad diffraction reflection in the 2θ range of $3 - 7^\circ$ (Fig. 4a).^{56, 57} In summary, we can reasonably infer that the defects in ML-UiO-66 membrane should be missing-linker defects.

Reference:

55. Feng, X.; Hajek, J.; Jena, H. S.; Wang, G.; Veerapandian, S. K. P.; Morent, R.; De Geyter, N.; Leyssens, K.; Hoffman, A. E. J.; Meynen, V.; Marquez, C.; De Vos, D. E.; Van Speybroeck, V.; Leus, K.; Van Der Voort, P., Engineering a highly defective stable UiO-66 with tunable lewis-bronsted acidity: the role of the hemilabile linker. *J. Am. Chem. Soc.* **2020**, 142 (6), 3174-3183.
56. Cliffe, M. J.; Wan, W.; Zou, X.; Chater, P. A.; Kleppe, A. K.; Tucker, M. G.; Wilhelm, H.; Funnell, N. P.; Coudert, F. X.; Goodwin, A. L., Correlated defect nanoregions in a metal-organic framework.

Nat. Commun. **2014**, *5*, 4176.

57. Liu, L.; Chen, Z.; Wang, J.; Zhang, D.; Zhu, Y.; Ling, S.; Huang, K. W.; Belmabkhout, Y.; Adil, K.; Zhang, Y.; Slater, B.; Eddaoudi, M.; Han, Y., Imaging defects and their evolution in a metal-organic framework at sub-unit-cell resolution. *Nat. Chem.* **2019**, *11* (7), 622-628.

R2-6. The authors compared their results with RO membranes, which is not fair. Please directly compare with PV or MD membranes. The performance data are moderate as compared to (1) Graphene oxide membrane for PV desalination (*J. Mater. Chem. A*, 2019, *7*, 18642–18652), in which the flux is $124 \text{ kg m}^{-2} \text{ h}^{-1}$; (2) PVDF membrane for MD desalination (*Journal of Membrane Science* 539 (2017) 34–42), in which the flux is $61.9 \text{ kg m}^{-2} \text{ h}^{-1}$.

Response: Thank you very much for kindly providing this valuable suggestion and the useful references. We agree with the reviewer's opinion that it is not fair to compare our PV performance with RO membranes. We acknowledge that in spite of higher flux we obtained compared with other polycrystalline membranes such as MOF and zeolite, the flux of UiO-66 membrane in our work is moderate when compared to other types of membranes such as PVA and graphene oxide (*via* PV process), and poly(vinylidene fluoride) (*via* MD process) due to the different nature of materials and membrane structures. However, we believe it is reasonable to compare performance based on same/similar membrane types (i.e., polycrystalline membranes such as MOF and zeolite) and membrane process (i.e., PV) in our work. To address the reviewer's concern, we have revised Fig. 3g (Page 11–12, revised manuscript) and included revisions (Line 205–207, Page 10; Line 251–257, Page 13, revised manuscript), which are as follows:

Line 205–207, Page 10, revised manuscript:

Even at 30°C (close to room temperature), a high water flux of $9.14 \pm 0.70 \text{ L m}^{-2} \text{ h}^{-1}$ was achieved with high rejection (~99.9%) for 35 g L^{-1} NaCl solution.

Line 251–257, Page 13, revised manuscript:

Moreover, besides excellent salt rejection, our ML-UiO-66 membrane exhibited a high water flux ($29.8 \text{ L m}^{-2} \text{ h}^{-1}$), outperforming other state-of-the-art polycrystalline zeolite and MOF membranes (Fig. 3g, Table S9). Nevertheless, it is moderate when compared with other types of membranes such as PVA and graphene oxide (*via* PV process)⁵², and poly(vinylidene fluoride) (*via* MD process)⁵³ due to the different nature of materials and membrane structures.

Fig. 3. Desalination performance of robust ultrathin UiO-66/ML-UiO-66 membranes. (a) Schematic diagram and (b) photograph of a PV setup for testing UiO-66/ML-UiO-66 membranes fabricated in this work. (c) Comparison of membrane thickness *via* a solution-based solvothermal method between UiO-66 membrane fabricated in the current study and other MOF membranes with lowest thickness reported in the literature.^{10, 21, 22, 27, 28, 33,}

⁴⁴⁻⁴⁸ **(d)** Water flux and salt rejection as a function of feed temperature during the PV process (35 g L^{-1} NaCl solution) (the inset shows an Arrhenius plot between water flux and reciprocal of feed absolute temperature). **(e)** Water flux and salt rejection as a function of feed salt concentration ranging from 5 g L^{-1} to 95 g L^{-1} at a constant feed temperature of 70°C . **(f)** Long-term operating desalination performance (water flux, salt rejection and permeate salt concentration) for the treatment of 35 g L^{-1} NaCl solution for 240 h (10 days) at 70°C without cleaning. (The red dashed line in Figs. 3f and 3h represents the limit ($\text{Na}^+ < 200 \text{ ppm}$) of drinking-water quality (fourth edition) issued by the World Health Organization (WHO)) **(g)** Comparison of water flux and rejection between our ultrathin ML-UiO-66 membrane and other state-of-the-art polycrystalline zeolite and MOF membranes via PV process (See details in Table S9, SI). **(h)** Long-term operating desalination performance (water flux, salt rejection and permeate salt concentration) for the treatment of hypersaline water (80 g L^{-1}) NaCl solution for 240 h (10 days) at 70°C without cleaning. **(i)** Water flux and salt rejection of ultrathin ML-UiO-66 membranes before and after immersion into NaClO solution (1000 ppm) for 240 h (10 days) for an accelerated chlorine resistance test (the inset shows the FE-SEM images of ML-UiO-66 membrane surface before **(a)** and after **(b)** the chlorine resistance test). (PV conditions: feed 35 g L^{-1} NaCl solution, temperature: 70°C) (Numbers that follows the \pm signs are standard deviation (SD) in this study.)

Reference:

52. Song, Y.; Li, R.; Pan, F.; He, Z.; Yang, H.; Li, Y.; Yang, L.; Wang, M.; Wang, H.; Jiang, Z., Ultrapermeable graphene oxide membranes with tunable interlayer distances via vein-like supramolecular dendrimers. *J. Mater. Chem. A*. **2019**, 7 (31), 18642-18652.
53. Lu, K.-J.; Zuo, J.; Chung, T.-S., Novel PVDF membranes comprising n-butylamine functionalized graphene oxide for direct contact membrane distillation. *J. Membr. Sci.* **2017**, 539, 34-42.

Response to Reviewer-3's comments

General Comments: Guiver and coworkers present a very exciting family of ultra-thin MOF membranes that exhibit exquisite water flux alongside requisite stability. They are of interest and may be suitable for publication in *Nature Materials* should the issues raised below be able to be adequately addressed:

Response: We appreciate the reviewer for kindly providing positive comments on the significance of our work. We have carefully considered all the comments/suggestions, given point-by-point responses (shown as follows), done additional experiments (Table S2, Fig. S9, Figs. S14–S15, Table S4–S5, Figs. S18–S20, Fig. S22 and Table S7) and made revisions to improve the quality of our manuscript and supplementary material.

R3-1. P6, Line 103-104. It mentioned, “Acetic acid (CH_3COOH) was used as a modulator to rationally quantitatively create missing-linker defects in ML-UiO-66 membranes”. The details of how Acetic acid can quantitatively modulate the defects should be included. Is it (the CH_3COOH , as a modulator) a general method that can work on other water-stable MOFs to form a missing-linker defects membrane?

Response: Thank you very much for this suggestion. During UiO-66 synthesis, acetic acid, as a modulator, coordinated with Zr_6 clusters *via* carboxyl group by competing with BDC linker molecules, resulting in the BDC-linker deficiency (i.e., missing-linker defects) in modulator-bearing secondary building units. Missing-linker defects could be quantitatively modulated by varying the acetic acid modulator concentration. Increasing acetic acid concentration would result in more missing-linker defects because more modulator molecules would compete with BDC linkers to form more modulator-bearing, instead of BDC-bearing, secondary building units in UiO-66 frameworks.

The modulation approach using acetic acid modulator is considered as a feasible method to create missing-linker defects in carboxylic-acid derived MOF membranes since acetic acid can coordinate with metal clusters by competing with carboxylic-

bearing linkers.³⁸ It would not be feasible to create missing-linker defects if other types of carboxyl-free linkers, such as imidazole, are used for MOF membrane synthesis.

To address the reviewer's concern, we have made revisions in the revised manuscript (Line 99–102, Page 5), which are shown as follows:

Acetic acid (CH₃COOH) was used as a modulator to rationally and quantitatively create missing-linker defects in ML-UiO-66 membranes by varying its concentration. It can coordinate with Zr₆ clusters *via* the carboxyl group by competing with BDC linker molecules.^{37, 38}

Reference:

37. Feng, Y.; Chen, Q.; Jiang, M.; Yao, J., Tailoring the Properties of UiO-66 through defect engineering: A review. *Ind. Eng. Chem. Res.* **2019**, *58* (38), 17646-17659.

38. Dissegna, S.; Epp, K.; Heinz, W. R.; Kieslich, G.; Fischer, R. A., Defective metal-organic frameworks. *Adv. Mater.* **2018**, *30* (37), e1704501.

R3-2. Figure 3 f and 3h, 10 days stability was demonstrated in these figures. Wondering how long the membrane performance can hold before lose its stability?

Response: Membrane stability is a relative concept, depending on many factors such as membrane properties, membrane structure and test conditions (such as temperature, salinity and other co-existing species). In our work, we performed a 10-day (240 h) stability test to preliminarily assess the operation performance of the UiO-66 membranes, which exhibited very stable performance for both water flux and salt rejection, even at operating times longer than most reported PV membranes (100 – 168 h) (Table S9, Pages S42–S43, revised supplementary material). Performance comparisons and defect-enhanced water transport mechanism are our key concerns in this manuscript, while longer-term stability is not our main concern, since we have already demonstrated stability over a time period well beyond most reports. We understand the reviewer's concern, and plan to longer-term performance when the resources are available.

R3-3. Figure 3g, comparisons with other MOFs/materials are made in both RO and PV scenarios. Curious about the experimental performance comparison between the control sample (without CH₃COOH modulator) and ML-UiO-66 using the same membrane fabrication method. Same for the performance in Figure 3 f and Figure 3h (comparison between the control sample and ML-UiO-66 in terms of stability in harsh conditions and salt rejection performance?). The comparison between the control and ML-UiO-66 in the Molecular Dynamics Simulations part is very clear.

Response: We sincerely appreciate the reviewer for providing these useful comments and suggestions. In our original supplementary material (Fig. S13, Page S24), we had provided experimental performance comparisons between the control sample UiO-66 (without CH₃COOH) and ML-UiO-66 membrane fabricated using the same method. To address the reviewer’s concern, we have also done additional stability experiments of UiO-66 membrane (the control sample) for comparison with ML-UiO-66 membranes (see Figs. S18 and S19, Page S35, revised supplementary material). An accelerated chlorine-resistance test was also conducted (Fig. S20, Page S36, revised supplementary material). We have also made revisions in the revised manuscript (see Line 244, Page 12; Line 259, Page 13; and Line 268, Page 13).

Fig. S18. Long-term operating desalination performance (water flux, salt rejection and permeate salt concentration) of UiO-66 membrane for the treatment of 35 g L⁻¹ NaCl solution for 240 h (10 days) at 70°C without cleaning. (The red dashed line represents the limit (Na⁺ < 200 ppm) of drinking-water quality (fourth edition) issued by the World Health Organization (WHO))

Fig. S19. Long-term operating desalination performance (water flux, salt rejection and permeate salt concentration) of UiO-66 membrane for the treatment of hypersaline water (80 g L^{-1}) NaCl solution for 240 h (10 days) at 70°C without cleaning. (The red dashed line represents the limit ($\text{Na}^+ < 200 \text{ ppm}$) of drinking-water quality (fourth edition) issued by the World Health Organization (WHO))

Fig. S20. Water flux and salt rejection of ultrathin UiO-66 membranes before and after immersion into NaClO solution (1000 ppm) for 240 h (10 days) for an accelerated chlorine-resistance test. The inset shows FE-SEM images of UiO-66 membrane surface before (a) and after (b) the chlorine-resistance test. (PV conditions: feed 35 g L^{-1} NaCl solution, temperature: 70°C)

R3-4. Simple explanation about Figure 4a will help the readers' understanding rather than just mention this figure. For example P16, Line 293-294: "In addition, missing-

cluster defects can be excluded according to low-angle experimental and simulated XRD patterns (Fig. 4a)".

Response: Thank you very much for this good suggestion! We have carefully checked the manuscript and made revisions with more explanations in the revised manuscript (Line 291–306, Page 15–16), which are as follows:

Compared to defect-free UiO-66, the existence of certain defects in the ML-UiO-66 structure is confirmed by increased specific surface area and lower linker weight loss based on BET and TG analysis (Figs. S24 and S25, Table S10). Generally, there are two types of intra-crystalline defects, which are missing-linker (Fig. 4g) and missing-cluster, in defective UiO-66 MOF structures. Compared with defect-free UiO-66 structure, a new broad XRD diffraction reflection is present in the 2θ range of $3 - 7^\circ$ only for missing-cluster defective UiO-66 structure due to the presence of *reo* phase.⁵⁵ In contrast, there is no such a characteristic diffraction reflection for both missing-linker defective and defect-free UiO-66 structures. Therefore, in our work, missing-cluster defects can be excluded according to the low-angle experimental and simulated XRD patterns in which there is no broad diffraction reflection in the 2θ range of $3 - 7^\circ$ (Fig. 4a).^{56, 57} In summary, we can reasonably infer that the defects in ML-UiO-66 membrane should be missing-linker defects. In a defect-free UiO-66 structure, each BDC²⁻ linker coordinates with two Zr₆ clusters. In contrast, for a missing-linker defect structure, the absence of one BDC linker with negative charge causes the formation of two defect centers having four coordinated unsaturated Zr sites.³⁷

Reference:

55. Feng, X.; Hajek, J.; Jena, H. S.; Wang, G.; Veerapandian, S. K. P.; Morent, R.; De Geyter, N.; Leyssens, K.; Hoffman, A. E. J.; Meynen, V.; Marquez, C.; De Vos, D. E.; Van Speybroeck, V.; Leus, K.; Van Der Voort, P., Engineering a highly defective stable UiO-66 with tunable lewis- bronsted acidity: the role of the hemilabile linker. *J. Am. Chem. Soc.* **2020**, *142* (6), 3174-3183.

56. Cliffe, M. J.; Wan, W.; Zou, X.; Chater, P. A.; Kleppe, A. K.; Tucker, M. G.; Wilhelm, H.; Funnell, N. P.; Coudert, F. X.; Goodwin, A. L., Correlated defect nanoregions in a metal-organic framework. *Nat. Commun.* **2014**, *5*, 4176.

57. Liu, L.; Chen, Z.; Wang, J.; Zhang, D.; Zhu, Y.; Ling, S.; Huang, K. W.; Belmabkhout, Y.; Adil, K.; Zhang, Y.; Slater, B.; Eddaoudi, M.; Han, Y., Imaging defects and their evolution in a metal-organic framework at sub-unit-cell resolution. *Nat. Chem.* **2019**, *11* (7), 622-628.

37. Feng, Y.; Chen, Q.; Jiang, M.; Yao, J., Tailoring the properties of UiO-66 through defect engineering: A review. *Ind. Eng. Chem. Res.* **2019**, *58* (38), 17646-17659.

R3-5. P16, Line 303-305: confused by the description. ~ 0.17% and ~ 1.09% are supposed for the defect-free samples rather than the missing-linkers UiO-66 density? But, missing-linker defect densities were also included here?

Response: The defect densities of ~ 0.17% and ~ 1.09% are both experimentally determined for the defect-free UiO-66 samples. Considering experimental errors, these values (~ 0.17%, ~ 1.09%) are indeed close to the theoretical value of zero. To clarify this issue, we have made revisions (Line 314–318, Page 16, revised manuscript), which are as follows:

Using the same protocols, the missing-linker defect densities of defect-free UiO-66 were experimentally determined to be only ~0.17% (BET, Fig. S24) and ~1.09% (TG, Fig. S25). Considering experimental errors, these values (~0.17%, ~1.09%) are indeed close to the theoretical value of zero, indicating its nearly defect-free nature, as expected.

R3-6. More details for terminal group confirmation for ML-UiO-66 in P17 lines 312-315? Comparison between defect-free and ML-UIO-66 in EDS (figure 4d) and FTIR (figure 4e); For figure 4e, is it supposed to be the comparison between the powder and membrane ML-UIO-66 rather than defect-free versus ML-UIO-66?

Response: Thank you for these helpful comments. We have first made revisions by discussing more details for the terminal group confirmation for ML-UiO-66 in the revised manuscript (Line 324–327, Page 16–17). We have also done additional EDS

and ATR-FTIR experiments, revised Figs. 4d and 4e, and revised the corresponding discussion (Line 327–338, Page 17, revised manuscript).

The revised figures and text are shown as follows:

Usually, the terminal group in defective UiO-66 is monocarboxylate, chloride, or hydroxy group, depending on the type of modulator.^{57, 59, 60} To further reveal the defect chemistry, compensating ligands were further identified in the ML-UiO-66 membrane structure. A broad band centered at 3450 cm^{-1} is due to the presence of intercrystallite water or/and physisorbed water (ATR-FTIR spectra, Fig. 4e).⁶² The carboxylate groups show characteristic bands at 1654 cm^{-1} (C=O stretching vibration), 1578 cm^{-1} (C=O antisymmetric stretching vibration) and 1398 cm^{-1} (C=O symmetric stretching vibration). The absence of additional hydroxy adsorption bands indicates that no hydroxy compensating ligand was formed under synthesis conditions where moisture did not interfere with the coordination reaction. Without the presence of chlorine element (EDS spectra, Fig. 4d) and additional hydroxy adsorption bands (ATR-FTIR spectra, Fig. 4e), we can conclude that the intra-crystalline defects in the ML-UiO-66 membrane structure are not terminated by chloride ligands or hydroxy groups.⁵⁹ Moreover, the liquid ^1H NMR spectrum indicates that the defect-compensating group is indeed monocarboxylate (Fig. 4f).

Fig. 4. Experimental characterization of missing-linker defects in ML-UiO-66 structure. **(a)** Low-angle ($2.6 - 20^\circ$, $\lambda = 1.541 \text{ \AA}$) experimental and simulated XRD patterns of UiO-66 structures with different intra-crystalline defects. **(b)** Simulated and experimental relationship between specific surface area and missing-linker defect density. **(c)** Defect density comparison between UiO-66 and ML-UiO-66 powders determined by Brunner–Emmet–Teller (BET) and thermogravimetric (TG) results (Figs. S24 and S25). **(d)** EDS spectra of UiO-66 and ML-UiO-66 membrane surface. **(e)** Attenuated total reflection-Fourier transform infrared spectroscopy (ATR-FTIR) spectra of UiO-66 and ML-UiO-66 membrane. **(f)** Liquid ^1H Nuclear magnetic resonance (NMR) spectrum (full chemical shift range) of ML-UiO-66 powder. **(g)** Structural representations of UiO-66 (defect-free) and ML-UiO-66 (missing-linker defect).

Reference:

57. Liu, L.; Chen, Z.; Wang, J.; Zhang, D.; Zhu, Y.; Ling, S.; Huang, K. W.; Belmabkhout, Y.; Adil, K.; Zhang, Y.; Slater, B.; Eddaoudi, M.; Han, Y., Imaging defects and their evolution in a metal-organic framework at sub-unit-cell resolution. *Nat. Chem.* **2019**, *11* (7), 622-628.
59. Shearer, G. C.; Chavan, S.; Bordiga, S.; Svelle, S.; Olsbye, U.; Lillerud, K. P., Defect engineering: tuning the porosity and composition of the metal-organic framework UiO-66 via modulated synthesis. *Chem. Mater.* **2016**, *28* (11), 3749-3761.
60. Shearer, G. C.; Chavan, S.; Ethiraj, J.; Vitillo, J. G.; Svelle, S.; Olsbye, U.; Lamberti, C.; Bordiga, S.; Lillerud, K. P., Tuned to perfection: ironing out the defects in metal-organic framework UiO-66. *Chem. Mater.* **2014**, *26* (14), 4068-4071.
62. Valenzano, L.; Civalleri, B.; Chavan, S.; Bordiga, S.; Nilsen, M. H.; Jakobsen, S.; Lillerud, K. P.; Lamberti, C., Disclosing the complex structure of UiO-66 metal organic framework: A synergic combination of experiment and theory. *Chem. Mater.* **2011**, *23* (7), 1700-1718.

R3-7. Maybe experimental Contact angle performance can be the further supplementary evidence for the hydrophilicity (water adsorption affinity) of ML-UiO-66 which was demonstrated in the Molecular Dynamics Simulations (Figure 5)?

Response: We appreciate the reviewer's suggestion. We had already provided the contact angle results in our original supplementary material (Fig. S15, Page S27).

We have revised the discussion of experimental contact angle results (Line 408-414, Page 21, revised manuscript), which is shown as follows:

Thus, we can conclude that the introduction of missing-linker defects increases water-to-membrane adsorption affinity (i.e., water adsorption ability, Figs. 5a and 5e), which is reflected by both enhanced membrane surface hydrophilicity (Fig. S21) and water uptake behavior (Fig. S22).⁶⁴ Moreover, it also endows faster diffusion transport of more water molecules *via* enlarging the pore window with a lower diffusion energy barrier of water molecules (Fig. S30) across sub-nanometer channels in the ML-UiO-66 membranes.

Reference:

64. Ghosh, P.; Colon, Y. J.; Snurr, R. Q., Water adsorption in UiO-66: the importance of defects.
Chem. Commun. **2014**, *50* (77), 11329-11331.

REVIEWER COMMENTS

Reviewer #1 (Remarks to the Author):

In my opinion, the authors responded exhaustively to all the reviewers' comments. The article is now ready for publication

Reviewer #2 (Remarks to the Author):

The authors have made substantial revisions, and the manuscript is ready for publication. One additional suggestion is that the authors may consider fluorescence confocal optical microscopy (FCOM) test to analyze the grain boundary.

Reviewer #3 (Remarks to the Author):

The authors have very thoroughly addressed the comments provided by the reviewers. In my view, the original manuscript was suitable for publication should the raised points be adequately addressed, and so having seen this occur, I am please to support the publication of the article and congratulate the authors on a very interesting contribution.

Point-by-point response to the reviewers' comments

Responses to Reviewer-1's comments

In my opinion, the authors responded exhaustively to all the reviewers' comments. The article is now ready for publication

Response: We appreciate very much your efforts on improving the quality of our manuscript.

Responses to Reviewer-2's comments

The authors have made substantial revisions, and the manuscript is ready for publication. One additional suggestion is that the authors may consider fluorescence confocal optical microscopy (FCOM) test to analyze the grain boundary.

Response: We appreciate very much your efforts on improving the quality of our manuscript. To address the reviewer's concern, we have conducted fluorescence confocal optical microscopy (FCOM) test to analyze grain boundary. In our experiments, the UiO-66 and ML-UiO-66 membranes with both ends sealed were dye-treated by dipping into 50 μM fluorescein ($\text{C}_{20}\text{H}_{12}\text{O}_5$, Aladdin) solution for 30 h. After drying, the FCOM test was conducted using an Olympus FV1000 confocal microscope.

The surface of both UiO-66 and ML-UiO-66 membranes was covered with fluorescein dye, but no fluorescence signal was detected inside the membranes ($\sim 5 \mu\text{m}$ below membrane surface). This indicates that there were no grain boundary defects in both membranes, which has been well confirmed by our gas selectivity results (Supplementary Table 4) as well as FE-SEM images (Fig. 2 and Supplementary Fig. 17).

Fig. 1 Fluorescence confocal optical microscopy (FCOM) images of UiO-66 (a, c) and ML-UiO-66 membranes (b, d). FCOM images of UiO-66 (a) and ML-UiO-66 (b) membrane surface. FCOM images of UiO-66 (c) and ML-UiO-66 (d) membrane slices ($\sim 5 \mu\text{m}$ below the membrane surface).

Responses to Reviewer-3's comments

The authors have very thoroughly addressed the comments provided by the reviewers. In my view, the original manuscript was suitable for publication should the raised points be adequately addressed, and so having seen this occur, I am please to support the publication of the article and congratulate the authors on a very interesting contribution.

Response: We appreciate very much your efforts on improving the quality of our manuscript.